

# Does who I am and what I feel determine what I see (or say)? A meta-analytic systematic review exploring the influence of real and perceived bodily state on spatial perception of the external environment

Erin MacIntyre, Felicity A. Braithwaite, Brendan Mouatt, Dianne Wilson and Tasha R. Stanton

IIMPACT in Health, Allied Health and Human Performance, University of South Australia, Adelaide, South Australia, Australia

## ABSTRACT

**Background:** Bodily state is theorised to play a role in perceptual scaling of the environment, whereby low bodily capacity shifts visuospatial perception, with distances appearing farther and hills steeper, and the opposite seen for high bodily capacity. This may play a protective role, where perceptual scaling discourages engaging with the environment when capacity is low.

**Methodology:** Our protocol was pre-registered *via* Open Science Framework (https://osf.io/6zya5/) with all amendments to the protocol tracked. We performed a systematic review and meta-analysis examining the role of bodily state/capacity on spatial perception measures of the environment. Databases (Medline, PsychINFO, Scopus, Embase, and Emcare) and grey literature were searched systematically, inclusive to 26/8/21. All studies were assessed using a customised Risk of Bias form. Standard mean differences and 95% CIs were calculated *via* meta-analysis using a random-effects model.

**Results:** A total of 8,034 studies were identified from the systematic search. Of these, 68 experiments (3,195 participants) met eligibility and were included in the review. These were grouped into the following categories: fatigue; pain; age; embodiment; body size/body paty size; glucose levels; fitness; and interoception, and interoceptive accuracy. We found low level evidence (limited studies, high risk of bias) for the effect of bodily state on spatial perception. There was consistent evidence that both glucose manipulations and age influence spatial perception of distances and hills in a hypothesised direction (lower capacity associated with increased distance and hill steepness). Mixed evidence exists for the influence of external loads, embodiment, body/body-part size manipulations, pain, and interoceptive accuracy. Evidence for fitness and/or fatigue influencing spatial perception was conflicting; notably, methodological flaws with fitness and fatigue paradigms and heterogenous spatial perception measures may underlie null/conflicting results.

**Conclusion:** We found limited evidence for bodily state influencing spatial perception of the environment. That all studies had high risk of bias makes conclusions about reported effects reflecting actual perceptual shifts (*vs* merely reflecting experimental demands or error due to inadequate study design)

Corresponding author
Tasha R. Stanton,
tasha.stanton@unisa.edu.au

pre-emptive. Rigorous evaluation is needed to determine whether reported effects reflect more than bias (*e.g.*, experimental demands, inadequate blinding). Future work using reliable measures of spatial perception, comprehensive evaluation of relevant confounders, and methodologically robust (and experimentally confirmed) bodily state experimental paradigms is warranted.

## INTRODUCTION

Historically, it was assumed that perception was veridical – that what one sees is an accurate representation of the external world. However, contemporary evidence challenges this assumption, instead suggesting that perception is an adaptive and emergent process constructed from complex and dynamic interactions between one's body and the environment (*Witt, 2020*). Inference models of perception posit that past-experiences and future action goals continuously reshape attention and perception (*Clark, 2013*; *Clark, 2018*). Here, perception is a hierarchical and generative process whereby top-down "predictions" aim to match (and therefore cancel out) incoming sensory input. Any residual "prediction error" continues upward for further processing. This flow of information is bidirectional, whereby predictions also flow downward to refine incoming sensory inputs. These predictions are theorised to be optimised by the free energy principle, where to maintain homeostasis, an organism must minimise the difference between its generative model and perceptual inferences in its interactions with the environment, and thus minimise surprise (*Friston, 2010*). In this model, action serves to minimise sensory prediction errors.

Embodied perception emphasises that perception is intrinsically related and grounded to one's sensory-motor capacity – perception is thus a bidirectional interaction of the person as a whole and their external environment (*Friston, 2011*). Action is key to many theories of perception, as movement and action are considered the interface between perception and the environment. Action both drives perception (looking in a direction determines what you see) and is driven by perception (turning your head towards a sound). Thus, action-specific accounts of perception, which frame perception of one's environment in terms of their ability to act within it, may give us insight into the mechanisms underlying perception. Based on the action-specific account, the Economy of Action (EoA) hypothesis proposes that perception acts like a "biological ruler", in that visual perception plays a role in scaling the external environment in line with an individual's body capacity (*Proffitt, 2006*). That is, hills look steeper when you are fatigued or carrying an external load (*Bhalla & Proffitt, 1999*) and distances appear longer if you have low blood glucose (*Cole & Balcetis, 2013*). Such perceptual shifts are thought to encourage the appropriate action given one's current bodily state/capacity (*e.g.*, explore *vs* exploit). By understanding the relationship between bodily state and visuospatial

perception of the environment, one can gain insight into ways through which perception may be scaled.

There is considerable criticism of the action-specific account of perception (*Durgin et al., 2009*; *Firestone, 2013*). Several studies have failed to replicate significant effects (*Durgin et al., 2009*; *Durgin et al., 2012*; *Hutchison & Loomis, 2014*), leading authors to conclude that initial results may be due to alternate theories. As perception itself cannot be directly measured, research relies on assessing *judgements*, and from these assessments infer the underlying perception (*Firestone, 2013*). However, as judgements are influenced by post-perceptual processes (*Firestone & Scholl, 2017*), the observed results may be due to influences other than physical demands. For instance, participant's responses may be influenced by knowledge effects (*e.g.*, an alpine skier estimating hill steepness) or by experimental demands (*e.g.*, a participant correctly guessing the hypothesis of the study and altering their response accordingly) (*Dean et al., 2016*; *Durgin et al., 2012*). Indeed, experiments that manipulate the participant's knowledge of experimental demands (*e.g.*, providing cover stories) have supported that social and contextual factors can influence visuospatial perceptual outcomes (*Keric & Sebanz, 2021*). This has led to repeated calls for robust experimental methodology to minimise such influences (*Firestone & Scholl, 2017*; *Witt, 2020*). However, little is known about how well current research has integrated these calls for improved rigour into study methodology.

A recent review (*Molto et al., 2020*) explored the effect of action-constraints (*i.e.*, specific to research evaluating external constraints, such as tool use) on visual perception and was unable to support the influence of experimental demand bias on measures of spatial perception. They specifically assessed whether experimental demand bias was present through moderator analysis of two commonly proposed sources of such bias – experimental design (within-group effects likely bigger than between-group effects) and type of spatial perception used (verbal measures likely to produce larger effects than visual or action-based measures). They found no evidence to support that either moderator contributed to experimental demand bias. However, such analysis may be pre-emptive, given that the authors did not critically appraise the overall risk of experimental bias of included studies. That is, null effects may reflect high heterogeneity of included studies induced by high levels of bias across many domains rather than evidence for or against experimental demand bias. Therefore, here we aimed to extend this past work by comprehensively exploring the effects of bodily state on spatial perception by including both modifiable *and* non-modifiable bodily states, and by formally evaluating methodological risk of bias. This risk of bias assessment is essential to extend pooled findings beyond simple replication to more nuanced interpretation. Thus, this systematic review and meta-analysis aims to summarise and critically appraise the current evidence for the effect of bodily state on measures of spatial perception of the environment.

## METHODOLOGY

Review methods were guided by the 2020 PRISMA guidelines (*Page et al., 2021*). The PRISMA 2020 checklist specific to our study has been completed and made available on Open Science Framework (OSF) (https://osf.io/6zya5/). This review was pre-registered

*via* uploading a locked protocol to OSF on 22/01/2021 (https://osf.io/6zya5/), prior to data extraction or analysis. All amendments to this protocol were tracked and uploaded to OSF.

## Eligibility criteria

There was no restriction on date or language of publication. For inclusion, studies were required to recruit adults (≥18 years), recruit a clinical or healthy population, and evaluate the effect of bodily state on an environmental spatial perception measure. Bodily state was considered to relate to physiological or morphological characteristics of participants. These states could be stable (*e.g.*, current body size), fluctuating (*e.g.*, blood glucose levels), or perceived (*e.g.*, self-rated fitness levels). States related to affect (*e.g.*, fear) or cognition (*e.g.*, executive function) were excluded. Study designs could be experimental or observational, using between- or within-group comparisons, or association designs; case studies were excluded. Spatial perception measures of the environment (*e.g.*, distance estimations or hill steepness estimations) were included, but measures of peri-personal space (including line bisection tasks to delineate boundaries between peri-personal and extra-personal space) and perception of object size were excluded. See Supplemental 1 for full details.

## Literature search

A systematic search of Medline (OVID), PsychINFO (OVID), Scopus (Elsevier), Embase (OVID), and Emcare (OVID) was undertaken from inception to 26/8/21. Keywords and subject headings related to both spatial perception and bodily states were used (Supplemental 2 provides the full search strategy).

Consistent with best-practice guidelines (*Higgins et al., 2021*), a grey literature search was conducted with non-peer reviewed studies considered eligible for inclusion in the review. Inclusion of grey literature was purposeful to help mitigate against publication bias and inflated effect sizes. Past work has shown that excluding grey literature can result in exaggerated effect sizes (*McAuley et al., 2000*). ProQuest Theses and Dissertations was searched using a similar search string, and Google Scholar was searched using ten strings of keywords, with the first 20 results of each search screened. Abstract titles from relevant conferences (Vision Sciences Society; European Conference on Visual Perception) were screened using keyword searches. Citations of all included studies were also screened for potentially relevant studies by the primary author.

## Study selection

Records identified by the search were exported to EndNote (Clarivate Analytics, Philadelphia, PA, USA) and duplicates were removed. The remaining records were uploaded to Covidence (*Veritas* Health Innovation Ltd., Melbourne, Australia) for screening. All title and abstracts were screened by two independent reviewers (EM, BM, FB, DW). Full texts of potentially relevant studies were then assessed for inclusion, again by two independent reviewers. Any disagreements were resolved through consensus discussion, and if needed, a third reviewer (TS) was consulted.

## Risk of bias assessment

Risk of bias was assessed using a customised, piloted tool based on the Appraisal tool for Cross-Sectional Studies (AXIS) (*Downes et al., 2016*), with modifications to allow appraisal of a range of study designs. The tool consisted of 15 items evaluating participant selection, study methodology, blinding, and reporting (See Supplemental File 3 for the full tool and scoring instructions). Two independent reviewers (EM, and BM, FB, or DW) appraised each study and discrepancies were resolved through discussion, and if required, *via* consultation with a third reviewer (TS).

## Data extraction

Two independent reviewers (EM and BM, FB, or DW) extracted the following data using a piloted, customised form: source details (authors, year); study design; study sample (bodily state examined, assessment method); participant information (sample size, age, and gender); presence of a control group/condition (if applicable); type of experimental manipulation (if applicable); the type of spatial perception measures used; manipulation check for bodily state (if applicable) and spatial perception measure results (measures of central tendency and dispersion or association). Discrepancies were resolved through discussion then, if required, through consultation with a third reviewer (TS). When outcome data were missing or reported in a form unamenable to quantitative meta-synthesis, authors were contacted a maximum of three times, after which the data were considered irretrievable (and included only in descriptive synthesis). In situations where data were presented only graphically and were not available from the authors, data were extracted by two independent reviewers (EM, and BM, FB, or DW) using WebPlotDigitizer (*Rohatgi, 2020*), which has high inter-coder reliability and validity (*Drevon, Fursa & Malcolm, 2017*). If >10% discrepancy between extractors occurred, data were extracted again by a single reviewer with the average of the two closest extractions used.

## Data analysis and synthesis

Studies/experiments were grouped into the following bodily state categories by the primary author: fatigue (sleep deprivation, pre/post exercise manipulations); pain; age; body/body part size manipulations; embodiment (ownership of a virtual body or a wheelchair); fitness; glucose manipulations; external loads (backpacks, ankle weights); and interoceptive accuracy. We considered such subgrouping necessary, as previous work (*Molto et al., 2020*) found that different constraint manipulations (tool-use, external loads, and effort) had different effects on measures of spatial perception (Hedge's g ranged from 0.14–0.4), which may represent different mechanisms through which spatial perception is scaled. Additionally, bodily state comparisons/manipulations had differing risks of bias associated with the comparison/manipulation. Given that studies with high risk of bias often have inflated effect estimates, subgrouping was necessary to avoid overall inflation of pooled estimates driven by certain bodily state comparisons/manipulations (*Schäfer & Schwarz, 2019*).
Our registered protocol proposed broad groupings of bodily state: (i) physiological, *e.g.*, glucose manipulation; (ii) morphological, *e.g.*, actual body weight/size; and (iii) perceived, *e.g.*, altered perception of the body without actual change, such as *via* virtual reality. While our chosen groupings are relatively consistent with the original protocol, the large number and variety of bodily state manipulations, with varying risk of bias of the manipulation, resulted in the need for further groupings. Given this deviation from original protocol, to mitigate confirmation bias, an assessor blinded to the study findings (TS) reviewed the groupings and made recommendations regarding pooling – *i.e.*, which outcome measures from included studies/experiments were sufficiently similar for quantitative synthesis.

## Meta-analysis method

Pooling was undertaken when data were available for at least two studies/experiments that were sufficiently similar (population and/or manipulation of bodily state, type of spatial perception measure used). Separate forest plots were completed for each spatial perception measure in each bodily state grouping, with subgroups based on the magnitude of spatial perception measures (*e.g.*, 4, 8 and 12 m for distance estimations). Studies/experiments using similar distances [+/−10%] were placed in the same subgroup. An overall pooled effect for all subgroups was considered when: (i) pooling did not contradict the study authors' hypotheses (*e.g.*, if an effect of short *vs* far distances was expected, overall pooling was not undertaken to avoid missing potential nuanced effects); (ii) there was sufficient variability of studies/experiments within each subgroup, with adequate sample sizes. If a study/experiment was included in numerous subgroups (*e.g.*, multiple distance estimations) and overall pooling was undertaken, the sample size was reduced based on the number of times it appeared in the forest plot. Overall pooling was avoided if it made data from small experiments meaningless due to reduced sample size.

Where sufficient data were provided, standard mean difference (Hedge's d) was calculated to provide a measure of effect size. For between-group designs the following formula was used (*Borenstein et al., 2011*).

$$d = \frac{\bar{X}_1 - \bar{X}_2}{S_{within}}$$

where $S_{within}$ is calculated by:

$$S_{within} = \sqrt{\frac{(n_1 - 1)SD_1^2 + (n_2 - 1)SD_2^2}{n_1 + n_2 - 2}}$$

For within-group designs Hedges *d* was calculated using the same formula, but using an appropriate calculation for $S_{within}$ to account for non-independence of the groups (*Borenstein et al., 2011*):

$$S_{within} = \frac{SD_{diff}}{\sqrt{2(1-r)}}$$

The variable $r$ is the correlation between the within subject data points. Where $r$ was not reported or could not be calculated with available data, we imputed a correlation value from a similar study (with similar comparisons); in these cases, sensitivity analyses were performed using correlations ±0.1, consistent with best practice (*Higgins et al., 2021*).

The majority of included studies ($N = 44$, 68%) had a sample size of less than 50 and the largest sample was 269. Therefore, because $d$ is biased in experiments with small sample sizes (*Hedges, 1981*), for consistency and ease of comparison, we computed Hedge's g for all studies by multiplying Hedge's d by correction factor J (*Borenstein et al., 2011*).

$$J = 1 - \frac{3}{4df - 1}$$

Standard error was calculated with the following formulas for between- and within-group designs respectively (*Borenstein et al., 2011*).

$$SE = \sqrt{\frac{n_1 + n_2}{n_1 \times n_2} + \frac{d^2}{2(n_1 + n_2)}}$$

$$SE = \sqrt{\left(\frac{1}{n} + \frac{d^2}{2n}\right) \times 2(1 - r)}$$

For experiments that evaluated the association between two continuous variables, Fisher's z and its variance were calculated from the correlation using the following formulas (*Borenstein et al., 2011*):

$$z = 0.5 \times \ln\left(\frac{1 + r}{1 - r}\right)$$

and

$$V_z = \frac{1}{n - 3}$$

Where experiments evaluating the same bodily state used differing outcomes (*e.g.*, correlations *vs* means), r was converted to d and the variance of d was calculated using the following formulas (*Borenstein et al., 2011*) to allow pooling, before being converted to g with the correction factor J.

$$d = \frac{2r}{\sqrt{1 - r^2}}$$

$$V_d = \frac{4V_r}{(1 - r^2)^3}$$

where $V_r$ was computed with the following formula:

$$V_r = \frac{(1 - r^2)^2}{n - 1}$$

When linear mixed models (LMM) were used and the original dataset was unavailable, the results were only presented narratively, consistent with recent recommendations (*Roth et al., 2018*). Specifically, given that effect sizes derived from LMM are dependent on the other factors included in the model, they should only be pooled when other studies also include the same factors within their analysis.

Calculated effect size and SE were imported to R (*R Development Core Team, 2021*) using the Metafor and Meta packages (*Balduzzi, Rücker & Schwarzer, 2019*; *Viechtbauer, 2010*). A random-effect, generic inverse variance meta-analysis was undertaken (*Higgins et al., 2021*). Where meta-analysis was not possible, the calculated effect size and 95% CIs are reported separately for each study. To assess potential for publication bias, contour enhanced funnel plots (*Egger et al., 1997*) were created for each forest plot, using the funnel.meta function, with contours set at $p < 0.1$, $p < 0.05$, and $p < 0.01$.

## RESULTS

### Search results

A total of 8,034 records were identified using the database search strategy. Of those, 2,536 were duplicates, resulting in 5,498 unique records screened for inclusion, with full text review of 94 records (containing 179 individual experiments). From this, 62 studies (132 experiments) were excluded (See Fig. 1), resulting in 32 studies (47 experiments) included. An additional 15 studies (21 experiments) were included through the grey literature search, resulting in a total of 47 studies (68 experiments). A list of all studies that underwent full text review and the reasons for their exclusion has been included in Supplemental File 4. Of these, 14 experiments had not been peer-reviewed (conference abstracts or dissertations). Forty-four experiments had sufficient data to allow potential inclusion in the quantitative analysis ($n = 25$ *via* graphical extraction). Additional data were obtained for 22 experiments *via* contacting the authors. In two experiments, data could not be obtained, thus effect sizes could not be calculated. Overall, 53 experiments (from a total of 37 studies) were included in the quantitative analysis. Table 1 provides methodological details of these included experiments.

### Characteristics of included experiments

Study design: The majority of experiments ($n = 42/68$) used between-group designs, either observational (*e.g.*, young *vs* old age groups), or *via* experimental manipulation (*e.g.*, glucose *vs* placebo glucose groups). Thirteen experiments used repeated measure within-group designs, and six experiments used mixed designs (within × within repeated measures; or between × within designs). Eight experiments used a cross-sectional design to explore associations between bodily state and spatial perception.

Spatial perception outcomes: All studies assessed either distance or hill/staircase steepness estimations. Estimated distances were to targets, or of aperture or gap size. Estimated distances ranged from 0.5–92.5 m, and estimated hill slopes from 2–39°. One study (*Tenhundfeld & Witt, 2015*) evaluated 'distance on hill' (difference between estimations for a distance on flat ground and the same distance on a hill). The methods

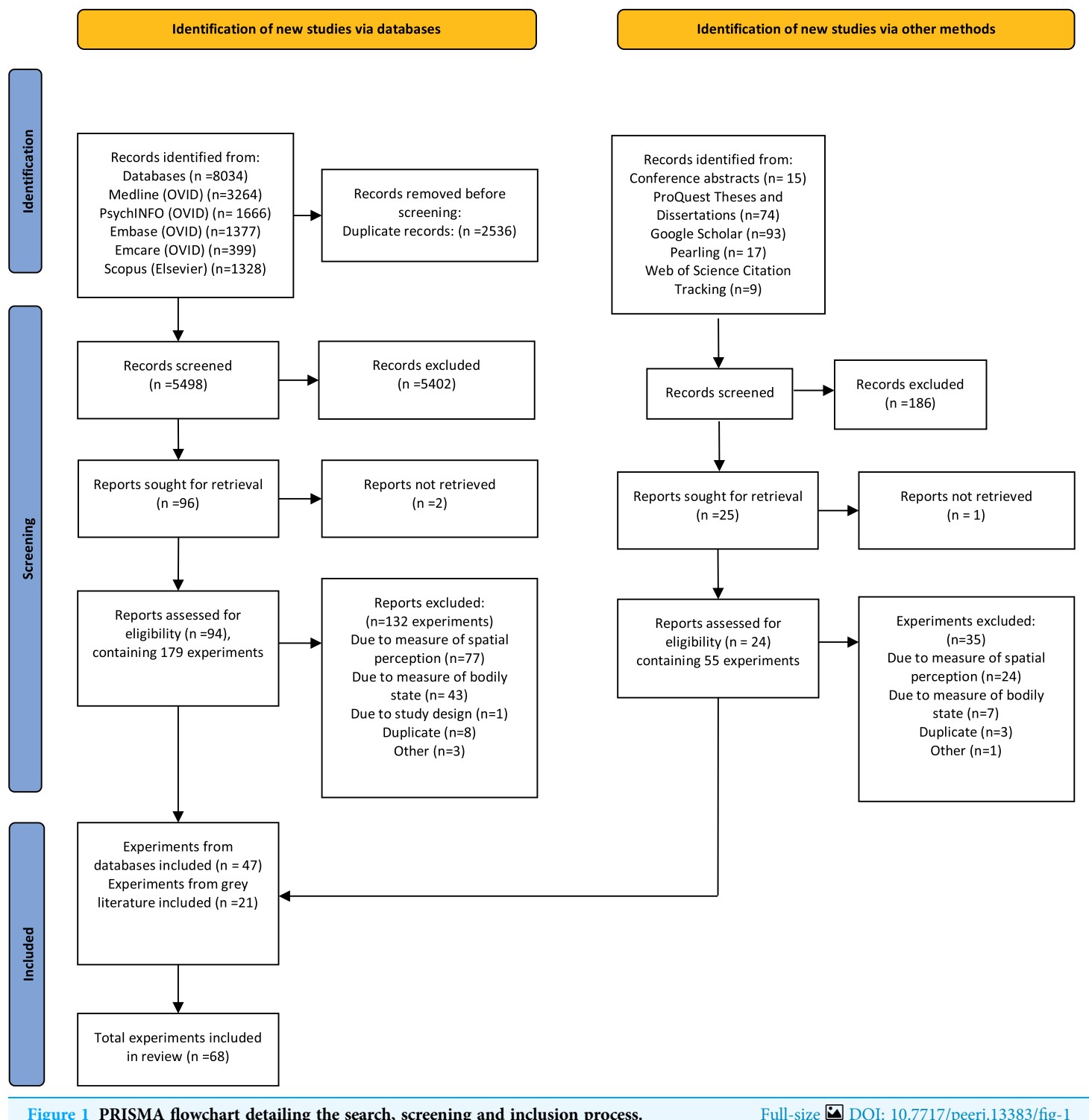

**Figure 1** **PRISMA flowchart detailing the search, screening and inclusion process.**

used to evaluate spatial perception were variable but were broadly categorized into verbal, visual (visual matching in the environment or on a diagram), haptic (participants used their hand to match the slope of the hill), or action-based (blind walking, beanbag toss).

**Table 1 Characteristics of included studies.**

| Grouping | Study details | | Study design | Participant details | Bodily state comparison, manipulation, and/or association | Spatial perception outcome |
|---|---|---|---|---|---|---|
| Age | Bhalla and Proffitt Experiment 4 | 1999 | Between group, cross sectional observational study | Older adults group $n = 32$, sex not reported; mean age 73 years, age range from 60–87 years (all >60 years of age) "Student aged" normative dataset $n = 300$, 150(f), age not reported The "student aged" group was previously collected as part of a normative dataset (Proffitt et al., 1995) | Older adults vs "student aged" cohort | Uphill steepness estimation Older adults judged four hills at two locations (hill set one were 2, 3, 4 and 25°, hill set two were 3, 5, 20 and 29°) Student aged group judged a single hill from a possible eight hills (2, 4, 5, 6, 10, 21, 31, 33 and 34°) Haptic, verbal, and visual matching measures |
| Age | Bian and Andersen Experiment 1 | 2012 | Between group, cross sectional observational study | Older adults group $n = 9$, 4(f); mean age 70.2 (4.9) years Young group $n = 8$, 4(f); mean age 22.8 (2.4) years | Older vs younger adults | Distance estimation at 4, 6, 8, 10 and 12 m Verbal Action based measure (blind rope pulling) |
| Age | Bian and Andersen Experiment 2 | 2012 | Between group, cross sectional observational study | Older adults group $n = 8$, 4(f); mean age 74.8 (5.1) years Younger group $n = 8$, 4(f); mean age 21.9 (2.3) | Older vs younger adults | Distance estimation at 4, 6, 8, 10 and 12 m Verbal |
| Age | Bian and Andersen Experiment 3 | 2012 | Between group, cross sectional observational study | Older adults group $n = 12$, 6 (f); mean age 76.2 (6.2) years Younger group $n = 12$, 6(f); mean age 23.2 (2.8) years | Older vs younger adults | Distance estimation at 4, 6, 8, 10 and 12 m Verbal |
| Age | Bian and Andersen Experiment 4 | 2012 | Between group, cross sectional observational study | Older adults group $n = 12$, 6 (f); mean age 71.8 (4.8) years Younger group $n = 12$, 6(f); mean age 23.2 (2.8) years | Older vs younger adults | Distance estimation at 4, 6, 8, 10 and 12 m Verbal |
| Age | Costello et al. Experiment 2 | 2015 | Between group, cross sectional observational study | Recruited $n = 114$. Excluded 8 participants as outliers, and 2 from older group for failing mini-mental state examination (<27) After exclusion older adults group $n = 52$, sex not reported; mean age 68.04 (range 60–80) years After exclusions younger group $n = 52$, sex not reported; mean age 22.40 (range 18–33) years | Older vs younger adults. Health status (multi-dimensional health assessment questionnaire), fatigue, and gait speed were also measured for both groups | Distance estimation at 3.4, 7.9, 13.4, 20.4 and 25.3 m Verbal |

| Grouping | Study details | | Study design | Participant details | Bodily state comparison, manipulation, and/or association | Spatial perception outcome |
|---|---|---|---|---|---|---|
| Age and Fitness | Dean et al. | 2016 | Exploratory cross-sectional study | $n = 106$ were recruited from the surrounding community. 60(f), age range from 18–72 years | Association between Age and hill steepness estimation Also investigated associations with: experiential (hill) knowledge; fitness (body mass index (BMI))**; personality traits; and sex | Uphill steepness estimation of three hills (9, 22.5 and 4.5°), always in the same order Verbal (all hills) Visual matching (9° hill only) Haptic (9° hill only) |
| Age | Eves et al. Experiment 1 | 2014 | Cross-sectional observational study | $n = 269$ participants were recruited at a train station. 129(f), mean age 38.1 years (SE = 0.87), age range from 18–84 years | Associations between age and stair steepness estimation | Stair steepness estimations, from the base of the stairs and 15 m away, the stairs were 23.4° steep Verbal, visual matching, and haptic measures |
| Age | Norman et al. | 2020 | Between group, cross sectional observational study | Older group $n = 16$, sex not reported; mean age 74.1 (SD = 6.1) years, age range from 65–83 years Younger group $n = 16$, sex not reported; mean age 21.3 (SD = 3.2) years, age range from 19–22 years | Older vs younger adults | Distance estimation using an equidistant cone task, where a cone was placed 6 m away and participants were then asked to place five cones to create five distances that are the same to the first interval Visual matching |
| Age | Sugovic and Witt Experiment 1 | 2013 | Between group, cross sectional observational study | Recruited $n = 29$, excluded 5 from analysis as failed vision test ($n = 4$ older group, $n = 1$ younger group) Post exclusion older group $n = 12$, 7(f); mean age 81.38 years Post exclusion younger group $n = 12$, 7(f); mean age 20.27 years | Older vs younger adults The Physical Activity Questionnaire was used as a measure of functional capacity | Distance estimation at 4, 6, 8 and 10 m Verbal |
| Body size | Bridgeman and Cooke Experiment 1 | 2015 | Between group, experimental design | Recruited $n = 32$. Excluded $n = 3$ (all increased eye height group), due to guessing hills to be 90° Increased eye height group (standing on a 37 cm box): $n = 17$, age and gender not reported Control group (standing on ground): $n = 15$, age and gender not reported | Eye height manipulated - participants stood on a 37 cm box at the base of a hill Mean eye height of this group is 187.9 cm (SD = 9.63 cm) compared to 140–160 cm for the control group | Uphill steepness estimation (12°) at four distances on the hill (2, 4, 8 and 16 m) Verbal estimates |

(Continued)

| Grouping | Study details | | Study design | Participant details | Bodily state comparison, manipulation, and/or association | Spatial perception outcome |
|---|---|---|---|---|---|---|
| Body size | Bridgeman and Cooke Experiment 2 | 2015 | Between group, experimental design. Replication of above with alternating allocation | Increased eye height group (standing on a 37cm box): $n = 16$, age and gender not reported Control group (standing on ground): $n = 17$, age and gender not reported | Eye height manipulated - participants stood on a 37 cm box at the base of a hill Mean eye height of this group is 187.9 cm (SD = 9.63 cm) compared to 140–160 cm for the control group | Uphill steepness estimation (12°) at four distances on the hill (2, 4, 8 and 16 m) Verbal estimates |
| Body size | Collier Chapter 5, Experiment1* | 2017 | Within group, experimental design | $n = 36$, 23(f), mean age 21.8 years | Hand size manipulated by wearing padded gloves. Each participant wore one padded, and one unpadded glove | Aperture width (4–14 cm) estimated using each hand. Visual matching task |
| Body size | Collier Chapter 5, Experiment 2* | 2017 | Within group, experimental design. Replication of above with addition of a cover story | $n = 36$, 23(f), mean age 25.9 years | Hand size manipulated by wearing padded gloves. Each participant wore one padded, and one unpadded glove | Aperture width (4–14 cm) estimated using each hand. Visual matching task |
| Body size | Jun et al. | 2015 | Between group, experimental design | Recruited $n = 42$, $n = 1$ excluded to having extremely large feet (>2 SD from mean), $n = 3$ due being outliers, and $n = 1$ due to missing data $n = 37$ analysed between both groups, no group sizes or demographics reported | Foot size was experimentally manipulated in virtual reality (VR). Feet rendered to be 51.9 cm and 12.97 cm in the large and small foot groups respectively | Distance estimations of a gap (0.5, 0.9, 1.05 and 1.7 m) Verbal estimates |
| Body size | van der Hoort et al. Experiment 9 | 2011 | Within group, experimental design. Replication of above with different measure of spatial perception | $n = 28$, 16(f), mean age 26.4 (SE = 1.2) years. All healthy adults naïve to the experimental hypothesis | Augmented reality (AR) "body swap" illusion, inducing embodiment of a large or small body | Distance estimation (8 m) Action based measure (blind walking) |
| Body size | van der Hoort et al. Experiment 10 | 2011 | Within group, experimental design | $n = 25$, 9(f), mean age 28.4 (SE = 1.5) years. All healthy adults naïve to the experimental hypothesis | AR "body swap" illusion, inducing embodiment of a large or small body | Distance estimation (4, 8, 16 m) Verbal estimations |
| Embodiment | Phillips et al.* | 2010 | Between group, experimental design | Recruited $n = 12$ for this study. 10 for avatar group and 2 additional male participants to add to a previous no avatar dataset Avatar group $n = 10$, 1(f), aged between 18–33 years. No avatar group $n = 10$, no demographics reported | Embodiment of an avatar vs no avatar in a VR environment | Distance estimations; random distance between 2.4–6.1 m Action based (blind walking) performance compared between VR and real world |

| Grouping | Study details | | Study design | Participant details | Bodily state comparison, manipulation, and/or association | Spatial perception outcome |
|---|---|---|---|---|---|---|
| Embodiment | Ries et al.* | 2008 | Between group, experimental design | Avatar group $n = 6$ and no avatar group $n = 5$. No demographics reported | Embodiment of an avatar vs no avatar in a VR environment | Distance estimations (length not reported) Action based (blind walking) performance compared between VR and real world |
| Embodiment | Scandola et al. Experiment 2 | 2019 | Within group, experimental design | $n = 20$, 2(f), mean age 44.2 (SD = 12.63). All had spinal cord injuries (SCI) and were manual wheelchair users (>6 months) | People with spinal cord injuries and use a wheelchair. Compared embodiment of own wheelchair vs other wheelchair | Distance estimation (2, 3, 4 m), verbal estimation; and uphill ramp steepness (4, 8, 16, 24, 32°); visual angles. Both in VR |
| External load | Bhalla and Proffitt Experiment 1 | 1999 | Between group, experimental design | Backpack group $n = 40$, 20(f), age not reported Control group $n = 90$, 45(f) The control group was previously collected as part of a normative dataset (*Proffitt et al., 1995*) | Experimental group wore a heavy backpack (~20% of body weight) Control group unencumbered | Uphill steepness measured at one of two hills (either 5° or 31°) Verbal, visual, and haptic measures |
| External load | Corlett et al. | 1990 | Between and within group, experimental design | Low resistance group $n = 8$ High resistance group $n = 8$ Overall mean age = 22 years | An external load (high or low resistance) was applied through an elastic band attached to the participant's belt Each group (high or low) underwent 8 conditions where the resistance was applied or not applied | Distance estimation (9 m) Action based measure (blind walking) |
| External load | Hutchinson and Loomis Experiment 1 | 2006 | Between group, experimental design | Backpack group $n = 12$, 6(f), age not reported Control group $n = 12$, 6(f), age not reported | Experimental group wore a heavy backpack (1/5–1/6 of body weight) Control group unencumbered | Distance estimation (3, 5, 7, 9, 11, 12 and 15 m) Verbal estimation Action based measure (blind walking) |
| External load | Hutchinson and Loomis Experiment 2 | 2006 | Within group, experimental design | $n = 12$, 8(f), age not reported | Experimental condition wore a backpack (1/5–1/6 of body weight) Control condition was unencumbered | Distance estimation (3, 8, 11 and 15 m) Verbal report |
| External load | Keric and Sebanz Experiment 3 | 2021 | Within group, experimental design | Recruited $n = 20$, 11(f), mean age 26.15 years Excluded $n = 3$, 2× due to guessing purpose of the study, 1× due to making attempts to measure distance | Compared wearing a backpack that was 20% of body weight to an empty backpack Cover story used | Distance estimation (every meter from 1–14 m) Verbal estimations |

(Continued)

| Grouping | Study details | | Study design | Participant details | Bodily state comparison, manipulation, and/or association | Spatial perception outcome |
|---|---|---|---|---|---|---|
| External load | Keric and Sebanz Experiment 4 | 2021 | Within group, experimental design | Recruited $n = 20$, 6(f), mean age 24.18 years Excluded 1× participant as they gave completely random estimates to finish quickly | Compared wearing a backpack that was 20% of body weight to an empty backpack No cover story used | Distance estimation (every meter from 1–14 m) Verbal estimations |
| External load | Lessard et al. | 2009 | Within group, experimental design | Recruited $n = 18$. Excluded $n = 2$ due to not following instructions Post exclusion $n = 16$, 8(f), age not reported | Experimental condition was ankle weights (5% of body weight) Control condition unencumbered | Distance estimations of gaps (0.15–3.05 m) Verbal estimation |
| External load | Proffitt et al. Experiment 1 | 2003 | Between group, experimental design | Backpack group $n = 12$, 7(f), age not reported Control group $n = 12$, 7(f), age not reported | Experimental group wore a heavy backpack (1/5–1/6 of body weight) Control group unencumbered | Distance estimation (4, 6, 8, 10, 12 and 14 m) Verbal estimation |
| External load | Shea and Masicampo | 2014 | 2 (backpack) × 2 (affirmation) factorial design | Total $n = 70$, 59(f), mean age 19.0 (SD = 1.07) n of each group and group demographics not reported | Experimental group wore a heavy (20% body weight up to 11 kg) backpack Control group wore and empty backpack | Distance estimation (39.6 m) Verbal estimation |
| External load | Vinson et al. Replication condition | 2017 | Within group, experimental design | $n = 12$, sex and age not reported | Experimental condition consisted of wearing a backpack that was 20% of body weight, followed by the control condition of no backpack | Distance estimation (8, 10, 12 and 14 m) Verbal estimation |
| External load | White Experiment 1a* | 2013 | Within group, experimental design | $n = 22$, 11(f), aged between 18–21 years | Compared two external load conditions (backpack manipulation and ankle weights both 10% of body weight) to a control condition (no external load) | Distance estimation (10, 12, 14 m) Reproduce a previously walked distance on a treadmill in VR environment |
| External load | White Experiment 1b* | 2013 | Within group, experimental design | $n = 26$, 9(f), aged between 18–26 years | Compared walking with two ankle weights conditions (5% and 10% of body weight) to a control condition (no external load) | Distance estimation (10, 12, 14 m) Reproduce a previously walked distance on a treadmill in VR environment |
| External load | White Experiment 2* | 2013 | Within group, experimental design | $n = 17$, 9(f), aged between 18–23 years | Compared three conditions of increased metabolic effort (condition of interest is ankle weights) to a control condition (normal walking) | Distance estimation (10, 12, 14 m) Reproduce a previously walked distance on a treadmill in VR environment |
| External load | White Experiment 4* | 2013 | Within group, experimental design | $n = 12$, 7(f), aged between 19–28 years | Compared three experimental conditions (condition of interest ankle weights) to two control conditions (eyes open and eyes closed) | Distance estimation (12 m) Blind-walking measure |

| Grouping | Study details | | Study design | Participant details | Bodily state comparison, manipulation, and/or association | Spatial perception outcome |
|---|---|---|---|---|---|---|
| Fatigue | Asaf et al.* | 2015 | Between (athletes vs non athletes), within group (stationary vs running on treadmill) experimental design | Athlete group $n = 15$ and non-athlete group $n = 15$ No other demographics reported | Compared performance of athletes and non-athletes before and during a fatiguing activity (treadmill running for 2 min) | Distance estimation (12, 18, 24 and 32 m) Visual matching |
| Fatigue | Baati et al. | 2020 | Within (normal sleep, first half of night, and second half of night sleep deprivation) × within (pre- and post-fatiguing exercise) experimental design | Recruited: $n = 10$, football players, 0(f), 22.8 (1.3) years | First half of night sleep deprivation (slept 3 am–7 am), second half of night sleep deprivation (slept 11 pm–3 am), compared to control night (slept 11 pm–7 am) Exercise condition was pre and post a repeated sprint cycling session | Distance estimation (15, 25 and 35 m) Verbal estimations |
| Fatigue | Baati et al. | 2015 | Within (sleep deprivation) × within (pre- and post-fatiguing exercise) experimental design | Recruited $n = 10$, football players, 0(f), 22.8 (1.3) years | Sleep deprivation condition (0 h sleep) compared to control night sleep (mean = 7 hr) Exercise condition was pre and post a repeated sprint cycling session | Distance estimation (15, 25 and 35 m) Verbal estimations |
| Fatigue | Bhalla and Proffitt Experiment 2 | 1999 | Within group, repeated measures | Recruited habitual runners, $n = 40$, 20(f); age not reported | Pre and post exhausting run, which was between 45–75 min long. Self-reported fatigue assessed | Uphill slope estimation of two hills (5° and 31°). Each participant estimated steepness of both hills, the order was counterbalanced. Verbal, visual matching, and haptic measures |
| Fatigue | Hunt et al. | 2017 | Between group experimental design | Opportunistic sample at a beach recruited $n = 83$ Excluded $n = 1$ (did not provide consent), final sample $n = 82$ f(45), mean age 53.63 (16.64) Randomly assigned to exercise ($n = 41$) and control ($n = 41$) groups. No group demographics reported | Exercise group underwent a 90 s session of exercise on a stepper machine immediately prior to distance estimation task. Control group did nothing Manipulation check measuring self-reported fatigue found the two groups were statistically different ($p = 0.002$) | Distance estimation (92.5 m) Verbal estimation |

(Continued)

| Grouping | Study details | | Study design | Participant details | Bodily state comparison, manipulation, and/or association | Spatial perception outcome |
|---|---|---|---|---|---|---|
| Fatigue | Jarraya et al. | 2013 | Between group (athletes, sedentary, control) × within group (pre- and post-exercise) design | Athletes were all members of a professional soccer team, $n = 18$, 0(f), mean age 23.7 years Sedentary group no participation in physical activity or sport, $n = 18$, 0(f), mean age 23.8 years Control group were "normally active", $n = 18$, 0 (f), mean age 22.7 years | All groups underwent an exercise test (2 min warm up, then 10 min of cycling at 30–50% maximal aerobic potential) Spatial perception measured before, after 1–2 min, 5–6 min, 9–10 min of exercise, and after the exercise test | Distance estimation (5, 7, 9 and 11 m) all done at each timepoint Verbal estimation |
| Fatigue | Proffitt et al. Experiment 5 | 1995 | Within group, repeated measures | Recruited $n = 60$ (30(f)) university students that were regular runners Excluded 4× due to hill steepness knowledge, and 3× students that weren't fatigued by the running task Final sample $n = 53$. No post-exclusion demographics provided | Pre and post exhausting run. No criteria on length or distance, just that participants needed to be fatigued at the end of the run | Uphill slope estimation of two hills (5° and 31°). Each participant estimated steepness of both hills, the order was counterbalanced. Verbal, visual matching, and haptic measures |
| Fatigue | Taylor-Covill and Eves Experiment 2 | 2013 | Between group experimental design | Fatigued group $n = 20$, 9(f), mean age 19.83 (SD = 1.58) years Control group $n = 20$, 9(f), mean age 20.41 (SD = 1.82) | Fatigued group participated in a maximal fitness test prior to spatial perception measure and control group did maximal fitness test after spatial perception measure | Slope estimation of a staircase (14.2°) was projected on a wall Verbal, visual, and haptic measures |
| Glucose | Cole and Balcetis Experiment 1 | 2013 | Between group, experimental design | Recruited $n = 54$ undergraduates, 40(f) Scarce bioenergetics (low blood glucose) group $n = 27$ Ample bioenergetics (high blood glucose) group $n = 27$ $n = 1$ participant excluded as outlier. Did not report which group this was from | "Bioenergetic" resources. Participants were given either a sugar drink or a placebo sugar drink. Blood glucose levels were measured after this manipulation | Distance estimation (length not reported) Action based measure (bean bag toss) |
| Glucose | Cole and Balcetis Experiment 2 | 2013 | Between group, experimental design | Recruited $n = 72$ undergraduates, 42(f), excluded $n = 2$ from ample group as they guessed that the drink was a placebo Scarce psychoenergetics $n = 35$ Ample psychoenergetics $n = 35$ | "Psychoenergetic" resources. Participants were all given non-caffeinated tea. The ample psychoenergetics group were told that the tea was a natural stimulant and the scarce psychoenergetics group were told that it was a sedative | Distance estimation (length not reported) Visual matching task |
| Glucose | Durgin et al. | 2012 | Between group, experimental design | Recruited $n = 39$ undergraduates, 20(f) Did not report the n of each group | Participants were either given a sugar drink or a placebo drink | Uphill steepness (8.6°) Verbal and haptic measures |

 

| Grouping | Study details | | Study design | Participant details | Bodily state comparison, manipulation, and/or association | Spatial perception outcome |
|---|---|---|---|---|---|---|
| Glucose | Schnall et al. Experiment 1 | 2010 | Between group, experimental design | 43 participants recruited, seven excluded from analysis (2× didn't follow instructions, 5× outliers) Did not report the n of each group | Participants were either given a sugar drink or a placebo drink. Manipulation check found that participants were no better than chance of guessing their group | Uphill steepness (29°) Verbal, visual, and haptic measures |
| Glucose | Zadra Experiment 1a* | 2013 | Between group, experimental design | n = 43, healthy participants, fasted for 4 h prior to experiment. Half were given a glucose drink, half a placebo drink. Sample size of each after exclusions (n = 6) not reported | Blood glucose levels were measured at baseline, after glucose/placebo drink and Stroop test, and after spatial perception task | Distance estimation (6–10 m in 0.5 m increments) Visual matching |
| Glucose | Zadra Experiment 3a* | 2013 | Between group, experimental design | Recruited n = 55, 30(f), mean age 18.92 (SD = 0.93) years. Divided into two groups: standard Jell-O (glucose) and placebo Jell-O (no glucose) Excluded 16 participants (1 due to procedural error, 1 due to drinking an extra drink during study, 1 for failure to follow instructions, 7 for previously participating in a slant study, and 6 as they ingested >10 g of Jell-O) | Oral glucose levels. Participants were instructed to chew, but not swallow, either standard Jell-O or placebo (no glucose) Jell-O Participants self-rated fitness was also measured | Uphill steepness (5.6°) Verbal and visual measures |
| Glucose | Zadra Experiment 3b* | 2013 | Between group, experimental design | Recruited n = 71, 42(f), mean age 18.68 (SD = 1.47) years. Dived into two groups: standard Jell-O (glucose) and placebo Jell-O (no glucose) Excluded 13 participants (2 due to extreme dislike of Jell-O, 3 for use of landmarks for distance estimation, 1 due to illness, 1 due to previously knowledge of study hypothesis, and 5 due to ingesting >10 g of Jell-O) | Oral glucose levels. Participants were instructed to chew, but not swallow, either standard Jell-O or placebo (no glucose) Jell-O Participants self-rated fitness was also measured | Distance estimation (6, 8, 10, 12 m) Visual matching (line bisection task) |

| Grouping | Study details | | Study design | Participant details | Bodily state comparison, manipulation, and/or association | Spatial perception outcome |
|---|---|---|---|---|---|---|
| Glucose Weight/ fitness | Schnall et al. Experiment 2 | 2010 | Between group, experimental design (glucose manipulation) Association study (self-reported measures) | 56 participants recruited, 10 excluded from analysis Did not report the n of each group | Glucose levels were measured before and after a glucose drink manipulation (sugar or placebo drink) Also used self-reported measures (bioenergetics test battery; self-reported sleep, exercise, nutrition, fatigue, mood, and stress) | Uphill steepness (5.6°) Verbal, visual, and haptic measures |
| Glucose and external load | Shaffer et al. | 2013 | 2 (backpack) × 2 (glucose) factorial design | four groups (total $n = 120$, 61 (f), no group demographics reported) placebo glucose, no backpack $n = 30$ placebo glucose, backpack $n = 30$ glucose, no backpack $n = 30$ glucose, backpack $n = 30$ | Glucose levels were not measured in this sample. However, they did report the results of a pilot study that found that the glucose blood levels were statistically different in a previous comparison of the glucose and placebo groups | Uphill steepness (16°) Verbal and haptic measures |
| Glucose | Zadra et al. | 2016 | Within (pre- and post-exercise) by within (glucose manipulation) experimental design | Recruited $n = 8$ participants, 3 (f), mean age 26.38 (SD = 5.53) years | Compared spatial perception measured pre- and post-exercise on two occasions, once having ingested a glucose supplement and once ingesting a placebo supplement (no glucose) | Distance estimation Action based (blind walking) |
| Interoception | Mouatt et al.* | 2021 | Association design | Recruited and analysed 20 participants, 10(f), mean age 30.2 (SD = 11.2) | Associations between interoceptive accuracy and uphill steepness estimation. Used a heart rate accuracy task. | Uphill steepness (5–15°) in virtual reality of 15 random hills Verbal estimates |
| Interoception | Tenhundfeld and Witt | 2015 | Between group, observational design | No details reported | Interoceptive accuracy - as measured by a HR accuracy task Split into high (2/3 of participants) and low (1/3 of participants) groups | Distance on hill |
| Pain | Tabor et al. | 2016 | Between group comparison, cross sectional | Recruited 72 participants (36 each group). Excluded $n = 10$ (5 each group) due to variable distance units used by participants Chronic pain group $n = 31$, 27(f); mean age 43.3 (SD = 11.1) years Healthy control group $n = 31$, 28(f); mean age 36.7 (SD = 13.7) years | Chronic pain group were all diagnosed with a chronic pain condition and recruited from a pain management centre. The average duration of pain was 12 (SD = 9.7) years and diagnosis were: back pain (50%), CRPS (9%), multi-site (41%) | Distance estimation at 4, 5, 7, 9 and 13 m Verbal estimates |

| Grouping | Study details | | Study design | Participant details | Bodily state comparison, manipulation, and/or association | Spatial perception outcome |
|---|---|---|---|---|---|---|
| Pain | Witt et al. | 2009 | Between group comparison, cross sectional | Recruited $n = 20$ participants (10 each group). Excluded 1× due to no pain during walking, and 2× due to being outliers<br>Chronic pain group $n = 8$, mean age 40.63 years<br>Control group $n = 8$, mean age 38.79 years | Chronic pain group consisted of people with MSK and neuropathic pain, average duration 9.02 (SD = 8.00) years. All self-reported pain with walking | Distance estimation at 4, 5, 7 and 9 m<br>Verbal estimates |
| Pain | Alaiti et al. | 2019 | Between group comparison, cross sectional | Painful shoulder group $n = 84$, 51(f); mean age 65(SD = 9.4) years<br>Control group $n = 51$, 35(f); mean age 57 (SD = 11.5) years | Chronic shoulder pain (right-handed and had pain in dominant shoulder for at least previous 3 months) | Distance estimation to a point 45–100 cm from body.<br>Visual matching (ratio judgement) |
| Weight/ fitness | Bhalla and Proffitt Experiment 3 | 1999 | Observational, between group design | total $n = 74$ university students recruited, 35(f) age not reported. Recruited 24 varsity athletes, 8(f) and 50 unfit people, 27(f) to ensure wide range of fitness and weight outcomes<br>seven participants excluded due to missing data (fitness assessments) | Fitness level via a cycling test. Used HR and estimated VO2 max to create a fitness score from -2–4<br>BMI scores | Uphill steepness estimation of four hills (4, 5, 21 and 31°)<br>Verbal, visual and haptic measures |
| Weight/ fitness | Cole et al. | 2013 | Observational, between group design | Total $n = 78$ participants recruited, 44(f), age not reported.<br>Split into low and high physiological potential groups<br>Did not report the n of each group | Fitness measured using waist-to-hip ratio (deviation from "gender-specific ideals". Participants with high (1SD above mean) and low (1SD below the mean) hip-to-waist ratios were categorized into the low and high physiological potential groups respectively | Distance estimation (4.87 m)<br>Visual matching (on paper diagram) |
| Weight/ fitness | Krpan and Schnall Experiment 1 | 2014 | Observational, between group design | Recruited $n = 54$, 25(f), mean age 28.15 (SD = 8.35). Excluded $n = 2$ due to failure to follow instructions<br>Split into low and high physical condition groups<br>Did not report the $n$ of each group | Self-rated fitness levels. Participants rated their current physical condition from 1 (very unwell) to 5 (excellent). Scores >1SD and <1SD from mean were the high and low fitness groups respectively | Uphill steepness estimation (39°)<br>Verbal, visual and haptic measures |

(Continued)
| Grouping | Study details | | Study design | Participant details | Bodily state comparison, manipulation, and/or association | Spatial perception outcome |
|---|---|---|---|---|---|---|
| Weight/ fitness | Krpan and Schnall Experiment 2 | 2014 | Observational, between group design | Recruited $n = 58$, 23(f), mean age 32.21 (SD = 12.18). Excluded $n = 1$ due to prior hill steepness knowledge Split into low and high physical condition groups Did not report the $n$ of each group | Self-rated fitness levels. Participants rated their current physical condition from 1 (very unwell) to 5 (excellent). Scores >1SD and <1SD from mean were the high and low fitness groups respectively | Uphill steepness estimation (39°) Verbal, visual and haptic measures |
| Weight/ fitness | Krpan and Schnall Experiment 3 | 2014 | Observational, between group design | Recruited $n = 150$, 52(f), mean age 27.6 (SD = 6.92) years. Excluded $n = 5$ for not completing all tasks, $n = 3$ for not complying with instructions, $n = 1$ for prior hill steepness knowledge Split into low and high physical condition groups Did not report the $n$ of each group | Self-rated fitness levels. Participants rated their current physical condition from 1 (very poor) to 6 (excellent). Scores >1SD and <1SD from mean were the high and low fitness groups respectively | Uphill steepness estimation (39°) Verbal, visual and haptic measures |
| Weight/ fitness | Shaffer et al. | 2019 | Association design | Recruited $n = 122$ (75f), with a median age of 18. They were separated into four experimental groups; however, they also performed a secondary analysis of the correlation between self-rated fitness and spatial perception measures Excluded $n = 2$ for steepness estimates over 90°, and $n = 6$ for guessing study hypothesis | Self-rated fitness levels Participants rated their physical fitness levels from 1 (not very physically fit) to 7 (very physically fit) | Uphill steepness estimation (6.1°) Verbal Distance estimation (cone on hill 10 m) Note - only done by 57 participants Verbal estimation |
| Weight/ fitness | Sugovic Experiment 2* | 2014 | Observational design, association study | Recruited $n = 73$, 40(f), mean age 32.4 (SD = 11.8) years. Excluded $n = 4$ due to being morbidly obese, $n = 2$ due to pregnancy, $n = 2$ due to "muscular body type", and $n = 1$ for being an outlier | Actual body size (BMI) and perceived body size (from modified Eating Disorders Questionnaire and figural scale) | Uphill steepness estimation (7°) Verbal (not reported as 41% of participants had difficulty) Visual |
| Weight/ fitness | Sugovic et al. | 2016 | Observational, between group design | Recruited $n = 66$, 30(f), mean age 24.4 (SD = 6.53) years. Excluded $n = 2$ due to not understanding task, $n = 2$ due to experimenter error, $n = 1$ as an outlier, $n = 1$ as one estimate was much further than others, and $n = 4$ for morbid obesity | Actual body size (BMI) grouped into normal (BMI 18.5–25), overweight (BMI 25–30), and obese (BMI 30–35) Perceived weight an evaluative measure of body size (too low - too high) and pictorial representation | Distance estimation (10, 15, 20 and 25 m) Verbal estimation |

| Grouping | Study details | | Study design | Participant details | Bodily state comparison, manipulation, and/or association | Spatial perception outcome |
|---|---|---|---|---|---|---|
| Weight/ fitness | Taylor | 2011 | Observational, between group design | Traceur (athlete) group $n = 27$, 0(f), mean age 19.9 years. Novices (control) group $n = 27$. 0(f), mean age 19.5 years Second control group $n = 18$, 0(f), mean age 20.1 years | Traceur group had a mean 16.77 months (SD = 15.4) experience in parkour Controls were matched on age, sex, and height | Distance estimation of wall heights (1.94, 2.29 and 3.45 m) Visual matching |
| Weight/ fitness | Taylor-Covill and Eves Experiment 1 | 2016 | Observational, between group design | Cross-sectional study recruited $n = 187$. Excluded $n = 7$ due to belief that all stairs are 45°, and $n = 9$ for failing angle knowledge tests Post exclusion mean age 41.8 (SD = 12.6), overweight group $n = 82$, 41(f) and healthy group $n = 89$, 42(f) | Experimenters used weight coding (BMI silhouettes) to separate into clearly healthy weight and clearly overweight groups | Staircase steepness (23.4°) Verbal, visual and haptic measures |
| Weight/ fitness | Taylor-Covill and Eves Experiment 2 | 2016 | Longitudinal cross-sectional study. Baseline assessment then a follow-up at 407 days (SD 137.45). | Recruited $n = 52$ at baseline, 41(f), mean age 37.4 (SD = 12.7) years Follow-up $n = 35$. Lost to follow up due to health problems ($n = 4$), pregnancy ($n = 2$), relocation ($n = 2$), and loss of contact ($n = 9$) Report that there were no statistical changes in demographics between the baseline and follow-up participants | Body composition measured using a Hologic "Discovery" dual-energy X-ray aborptiometry body scanner | four Staircase steepness estimations (between 20–33°) Verbal, visual and haptic measures |

**Notes:**
\* Experiments that have not been peer reviewed.
\*\* Experiments that contain results that are relevant to more than one group.
All reported SS is based on that which was analysed unless otherwise stated.

## Overall risk of bias

All experiments had high risk of bias (Table 2). No experiments used random sampling and only 4% ($n = 3/68$) performed *a priori* sample size calculations (*Scandola et al., 2019*; *Shaffer, Greer & Schaffer, 2019*). Four experiments (*Dean et al., 2016*; *Durgin et al., 2012*; *Shaffer et al., 2013*; *Taylor-Covill & Eves, 2013*) had low risk of bias for the spatial perception measure (*i.e.*, reliability established by previous research). Only 22% ($n = 15/68$) of experiments adequately controlled for confounding variables (such as controlling for time of day or previous night's sleep in fatigue paradigms). Blinding was poorly applied: 21% ($n = 14/68$) had a low risk of bias for blinding participants (participants were blinded and this was assessed with post-experiment questioning), 3% ($n = 2/68$) blinded assessors, and no experiments reported a blinded analysis. None of the

MacIntyre et al. (2022), PeerJ, DOI 10.7717/peerj.13383

**Table 2 Risk of bias for included studies.**

| Study | Selection | | Methodological | | | | | Blinding | | | Reporting | | | | Other |
|---|---|---|---|---|---|---|---|---|---|---|---|---|---|---|---|
| | Selection method | Sample size | Group Randomised | Task Randomised | Confounding variables | Manipulation achieved | Reliability | Participant blinding | Assessor blinding | Analysis blinding | Hypothesis a priori | Analysis a priori | Outcomes reported | Missing data <15% | |
| Alaiti et al. | + | + | N.A. | — | ? | — | ? | ? | ? | ? | + | ? | — | ? | |
| Asaf et al. | + | ? | + | — | ? | — | ? | ? | + | ? | ? | + | + | + | |
| Baati et al. | + | ? | — | — | + | — | ? | ? | + | ? | + | ? | — | ? | |
| Baati et al. | + | ? | — | + | + | — | ? | ? | + | ? | + | ? | — | — | |
| Bhalla and Proffitt Experiment 1 | + | ? | + | ? | + | — | + | + | + | ? | + | ? | — | ? | |
| Bhalla and Proffitt Experiment 2 | + | ? | N.A. | — | + | + | + | + | + | ? | + | ? | — | ? | |
| Bhalla and Proffitt Experiment 3 | + | ? | N.A. | — | ? | — | + | ? | + | ? | + | ? | — | — | |
| Bhalla and Proffitt Experiment 4 | + | ? | N.A. | — | + | + | ? | ? | + | ? | + | ? | — | ? | |
| Bian and Andersen Experiment 1 | + | ? | N.A. | — | + | — | ? | + | + | ? | ? | ? | — | — | |
| Bian and Andersen Experiment 2 | + | ? | N.A. | — | + | — | + | + | + | ? | ? | ? | — | — | |
| Bian and Andersen Experiment 3 | + | ? | N.A. | — | + | — | + | + | + | ? | ? | ? | — | — | |
| Bian and Andersen Experiment 4 | + | ? | N.A. | — | + | — | + | + | + | ? | ? | ? | — | — | |
| Bridgeman and Cooke Experiment 1 | + | ? | + | — | ? | — | ? | + | + | ? | + | ? | — | — | 1 |
| Bridgeman and Cooke Experiment 2 | + | ? | + | + | ? | — | + | ? | + | ? | + | ? | — | — | 1 |
| Cole and Balcetis 2013 Experiment 1 | + | ? | ? | N.A. | + | — | + | ? | + | ? | + | ? | — | — | 2 |
| Cole and Balcetis 2013 Experiment 2 | + | ? | — | N.A. | + | — | + | — | — | ? | + | ? | — | — | |
| Cole et al. | + | ? | ? | N.A. | ? | — | ? | — | ? | ? | + | ? | — | — | |

MacIntyre et al. (2022), *PeerJ*, DOI 10.7717/peerj.13383

| Study | Selection | | Methodological | | | | | Blinding | | | Reporting | | | | Other |
|---|---|---|---|---|---|---|---|---|---|---|---|---|---|---|---|
| | Selection method | Sample size | Group Randomised | Task Randomised | Confounding variables | Manipulation achieved | Reliability | Participant blinding | Assessor blinding | Analysis blinding | Hypothesis a priori | Analysis a priori | Outcomes reported | Missing data <15% | |
| Collier Chapter 5, Experiment 1 | + | ? | ? | — | + | — | + | ? | + | ? | + | ? | — | — | |
| Collier Chapter 5, Experiment 2 | + | ? | ? | — | + | — | + | + | + | ? | + | ? | — | — | |
| Corlett et al. | + | ? | — | + | ? | + | ? | ? | ? | ? | ? | ? | — | ? | 3 |
| Costello et al. Experiment 2 | + | ? | N.A. | + | + | — | + | ? | + | ? | + | ? | — | — | |
| Dean et al. | + | ? | N.A. | + | — | — | — | + | + | ? | ? | ? | — | — | 4 |
| Durgin et al. | + | + | ? | ? | + | + | — | — | + | ? | + | + | + | ? | 4 |
| Eves et al. Experiment 1 | + | ? | N.A. | + | — | — | + | + | + | ? | + | ? | — | ? | |
| Hunt et al. | + | + | — | N.A. | — | — | ? | ? | ? | ? | + | ? | — | ? | |
| Hutchinson and Loomis Experiment 1 | + | ? | + | + | ? | — | + | + | + | ? | + | ? | — | — | |
| Hutchinson and Loomis Experiment 2 | + | ? | + | — | ? | — | + | + | + | ? | + | ? | — | — | |
| Jarraya et al. | + | ? | N.A. | — | + | — | ? | ? | + | ? | + | + | ? | ? | |
| Jun et al. | + | ? | — | + | — | — | ? | + | ? | ? | + | + | — | — | |
| Keric and Sebanz Experiment 3 | + | ? | + | — | + | — | ? | — | ? | ? | + | ? | — | — | |
| Keric and Sebanz Experiment 4 | + | ? | + | — | + | — | ? | + | ? | ? | + | ? | — | — | |
| Krpan and Schnall Experiment 1 | + | ? | N.A. | — | + | + | ? | + | + | ? | + | + | — | — | |
| Krpan and Schnall Experiment 2 | + | ? | N.A. | — | + | + | + | — | + | ? | + | ? | — | — | |
| Krpan and Schnall Experiment 3 | + | ? | N.A. | — | + | + | + | — | + | ? | + | ? | — | — | |
| Lessard et al. | + | ? | + | — | — | — | ? | ? | + | ? | + | ? | — | — | |
| Mouatt et al. | + | — | N.A. | — | + | — | + | — | + | ? | ? | ? | — | — | |
| Norman et al. | + | ? | N.A. | N.A. | ? | — | + | + | + | ? | + | ? | — | — | |
| Phillips et al. | + | ? | + | ? | ? | ? | ? | ? | + | ? | + | ? | — | — | |

(Continued)

| Study | Selection | | Methodological | | | | | Blinding | | | Reporting | | | | Other |
|---|---|---|---|---|---|---|---|---|---|---|---|---|---|---|---|
| | Selection method | Sample size | Group Randomised | Task Randomised | Confounding variables | Manipulation achieved | Reliability | Participant blinding | Assessor blinding | Analysis blinding | Hypothesis a priori | Analysis a priori | Outcomes reported | Missing data <15% | |
| Proffitt et al. Experiment 1 | + | ? | + | — | + | — | ? | + | + | ? | + | ? | — | ? | |
| Proffitt et al. Experiment 5 | + | ? | N.A. | — | ? | + | ? | + | + | ? | + | ? | — | — | |
| Ries et al. | + | ? | + | ? | ? | — | ? | + | + | ? | + | ? | — | — | |
| Scandola et al. Experiment 2 | + | — | N.A. | ? | ? | — | ? | ? | + | ? | + | ? | — | — | 5 |
| Schnall et al. Experiment 1 | + | ? | ? | — | — | + | ? | — | ? | ? | + | ? | — | + | |
| Schnall et al. Experiment 2 | + | ? | ? | — | — | — | + | — | ? | ? | + | ? | — | + | |
| Shaffer et al. 2013 | + | ? | ? | ? | + | — | — | — | + | ? | + | ? | + | ? | 4 |
| Shaffer et al. 2019 | + | — | — | — | ? | — | ? | — | + | ? | + | ? | — | — | |
| Shea and Masicampo | + | ? | — | N.A. | + | — | + | + | + | ? | + | + | — | ? | |
| Sugovic Experiment 2 | + | ? | N.A. | ? | ? | — | + | ? | + | ? | ? | ? | + | + | |
| Sugovic and Witt Experiment 1 | + | ? | N.A. | — | ? | — | ? | ? | + | ? | + | ? | — | + | |
| Sugovic et al. | + | ? | N.A. | — | ? | — | ? | ? | + | ? | ? | ? | — | — | |
| Tabor et al. | + | ? | N.A. | — | + | — | ? | ? | + | ? | + | ? | — | — | |
| Taylor et al. | + | ? | N.A. | + | — | ? | ? | ? | + | ? | ? | ? | — | ? | |
| Taylor—Covill and Eves 2013 Experiment 2 | + | ? | + | N.A. | + | ? | — | + | + | ? | + | ? | — | ? | |
| Taylor—Covill and Eves 2016 Experiment 1 | + | ? | N.A. | — | + | + | + | + | + | ? | ? | ? | — | — | |
| Taylor—Covill and Eves 2016 Experiment 2 | + | ? | N.A. | — | — | — | + | + | + | ? | ? | ? | — | + | |
| Tenhundfeld and Witt | ? | ? | N.A. | ? | ? | + | ? | ? | + | ? | ? | ? | + | ? | 6 |
| van der Hoort et al. Experiment 10 | ? | ? | ? | N.A. | — | ? | ? | ? | + | ? | + | ? | + | — | |
| van der Hoort et al. Experiment 9 | ? | ? | — | — | — | ? | ? | ? | + | ? | + | ? | + | — | |

| Study | Selection | | Methodological | | | | | Blinding | | | Reporting | | | | Other |
|---|---|---|---|---|---|---|---|---|---|---|---|---|---|---|---|
| | Selection method | Sample size | Group Randomised | Task Randomised | Confounding variables | Manipulation achieved | Reliability | Participant blinding | Assessor blinding | Analysis blinding | Hypothesis a priori | Analysis a priori | Outcomes reported | Missing data <15% | |
| Vinson *et al.* | + | + | + | ? | ? | — | + | — | + | ? | ? | ? | + | ? | |
| White Experiment 1a | ? | ? | — | — | ? | — | ? | ? | + | ? | + | ? | — | ? | 7 |
| White Experiment 1b | ? | ? | — | — | ? | — | ? | ? | + | ? | + | ? | — | ? | 7 |
| White Experiment 2 | ? | ? | — | — | ? | — | ? | ? | + | ? | + | ? | — | ? | 7 |
| White Experiment 4 | ? | ? | — | — | ? | — | ? | ? | + | ? | + | ? | — | ? | 7 |
| Witt *et al.* | + | ? | N.A. | — | ? | — | ? | + | ? | ? | ? | ? | — | + | |
| Zandra Experiment 1a | + | ? | — | — | — | — | ? | + | — | ? | ? | ? | + | — | |
| Zandra Experiment 3a | + | ? | ? | N.A. | — | — | ? | — | ? | ? | ? | ? | + | + | |
| Zandra Experiment 3b | + | ? | ? | — | — | — | ? | — | ? | ? | ? | ? | — | + | |
| Zandra *et al.* | + | ? | — | — | — | — | ? | + | ? | ? | + | ? | + | + | |

**Notes:**

For each category, — represents low risk of bias, + represents high risk of bias, ? represents unclear risk of bias, and N.A., represents Not Applicable.

Details for Other Risk of Bias: (1) Risk of response bias as participants were repeatedly asked to estimate the same slope; (2) Likely poor reliability of spatial perception measured with action-based beanbag toss; (3) Manipulation of bodily state through experimenters adding external load via elastic bands, unlikely to be consistent within/between participants; (4) Scored low on reliability, however only one of the three measures of hill steepness has been assessed for reliability; (5) 8.9% of slope data contained estimations >90°; (6) Primary author reported in correspondence that the group attempted to replicate the effect and failed (and these data have not been published –publication bias); (7) Potential bias in measurement of spatial outcome by assessor (unblinded assessor used a stopwatch to calculate distance walked).

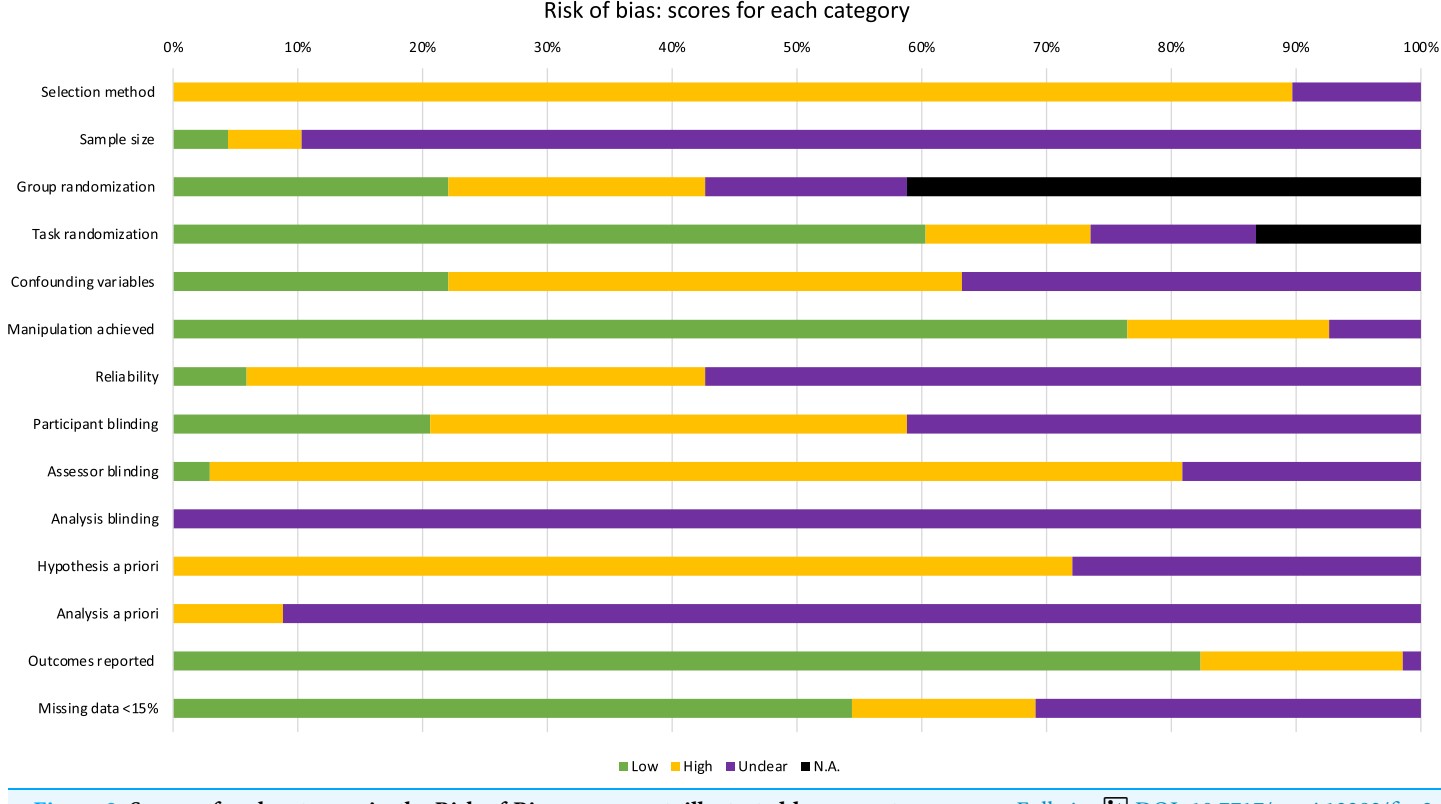

**Figure 2** Scores of each category in the Risk of Bias assessment, illustrated by percentage.

included experiments were pre-registered. Figure 2 illustrates the percentage of experiments that scored high, low, unclear, or N/A for each risk domains.

## Outcomes: effect of bodily state on spatial perception

### Effect of fatigue on spatial perception (n = 8)

All experiments aimed to induce fatigue through exercise, comparing measures of spatial perception either pre-and post-exercise, or between exercise and non-exercise groups. Two experiments used a fatigue manipulation *via* sleep deprivation in addition to an exercise manipulation.

### Distance estimation (n = 5)

All experiments evaluated verbal distance estimation data. Pooling of two experiments (*Baati et al., 2020*; *Baati et al., 2015*) that compared distance perception pre- *vs* post-exercise using a sprint cycling task (10 × 6 s maximal cycling) found no effect of exercise at any of the three distances measured, for either the normal (Fig. 3A) or the sleep deprived conditions (Fig. 3B). Within the same studies, there was an effect of fatigue due to sleep deprivation only at the 35 m distance (Fig. 3C), where participants perceived the distance as *closer* when undergoing sleep deprivation compared to a normal night sleep. A final study (*Hunt, Hunt & Park, 2017*) evaluated the effect of fatigue (stepper machine 90 s) *vs* non-fatigue, finding that the fatigued group estimated a single target distance of

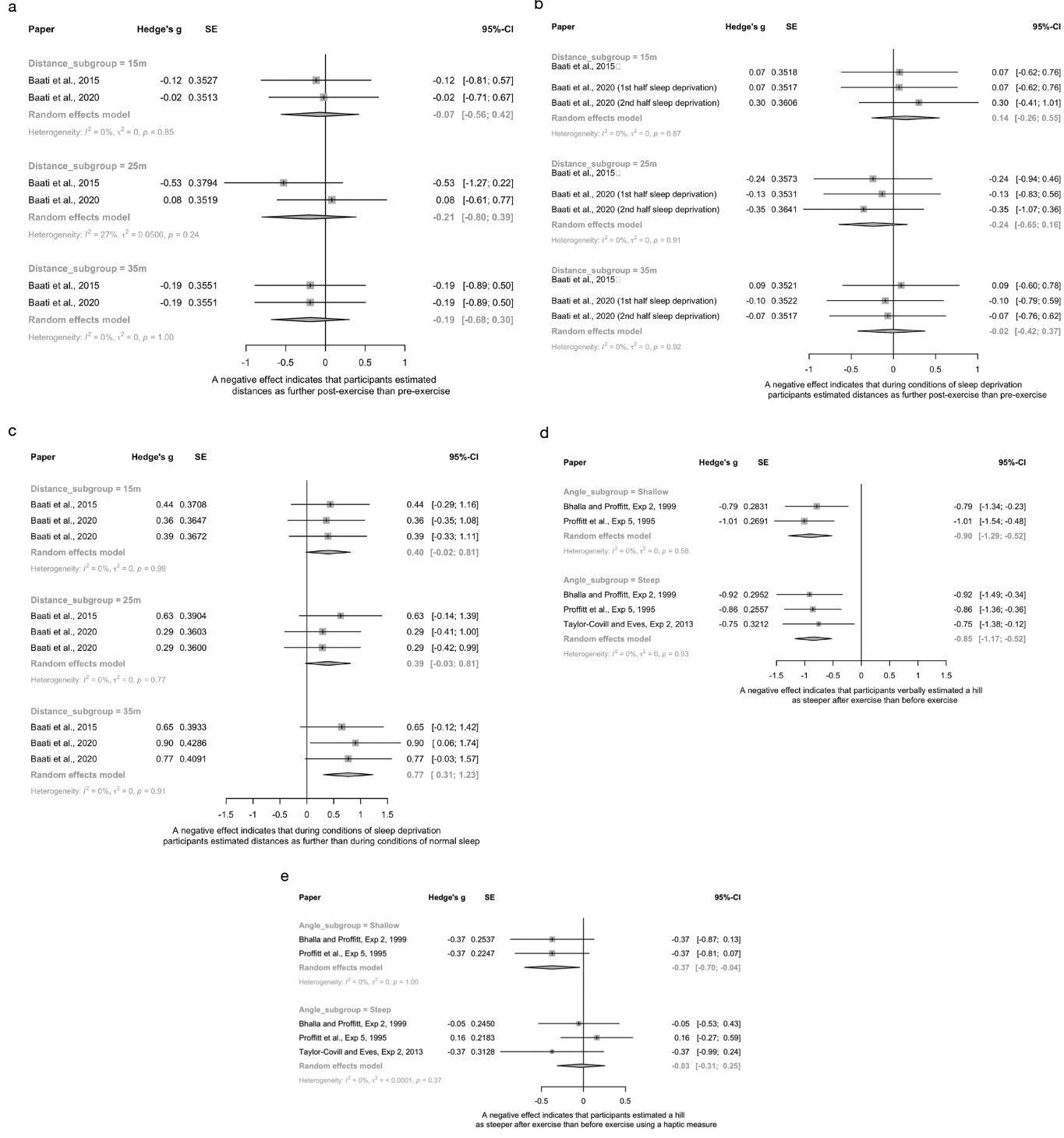

**Figure 3 Pooled estimates for the effect of fatigue on spatial perception.** (A) Verbal distance estimation pre- and post-exercise, subgrouped by distance. (B) Verbal distance estimation pre- and post-exercise during conditions of sleep deprivation, subgrouped by distance. (C) Verbal distance estimation in conditions of normal sleep and sleep deprivation, subgrouped by distance. (D) Verbal hill steepness estimations pre- and post-exercise, subgrouped by hill steepness. (E) Haptic estimations of hill steepness pre- and post-exercise, subgrouped by hill steepness. In all cases a negative effect indicates that the fatigued group estimated spatial perception measures (distance or steepness) as larger than the control group.

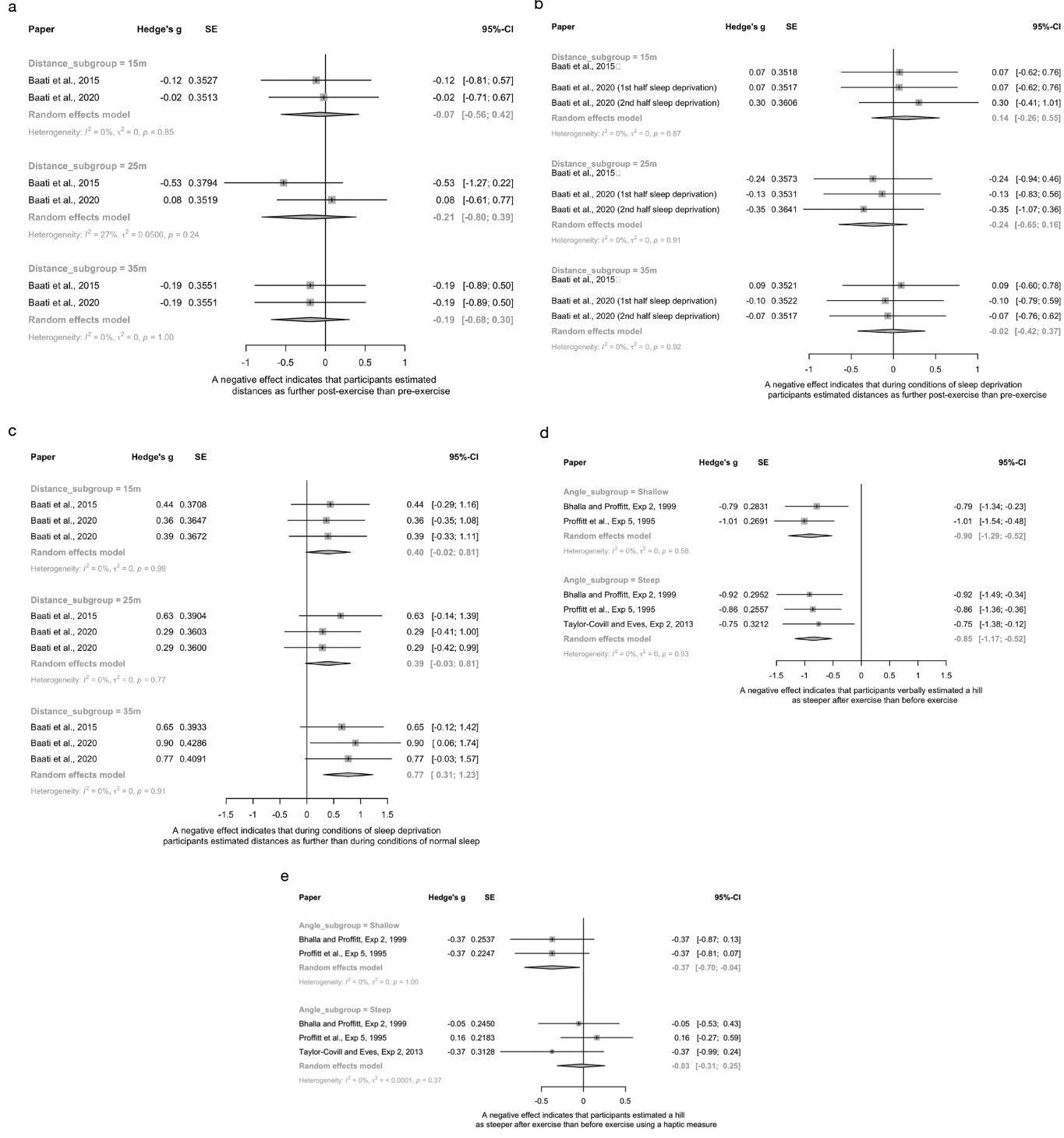

92.5 m as being farther than the non-fatigued group (Hedge's g: 0.56, 95% CI [−0.95 to −0.16]).

Two experiments used comparable distance measures, but in athletic populations (*Asaf, Santillán & Barraza, 2015*; *Jarraya et al., 2013*). *Asaf, Santillán & Barraza (2015)* showing no difference (Hedge's g range: −0.13 to 0.09) in distance estimations based on their fatigue manipulation (treadmill running 2 min), whereas *Jarraya et al. (2013)* found very large effect sizes following cycling for 10 mins (Hedge's g range: 8.32 to 10.08), indicating participants estimated distances as *closer* in the post-exercise (fatigued) condition compared to the pre-exercise condition. Pooling of the studies' similar distances (12 m and 11 m) was non-significant, with high heterogeneity ($I^2$ = 96%). Large differences in effect sizes at a similar distance (g = −0.14 *vs* 9.92), suggest that between study differences are unlikely driven by differences in the spatial perception task; but rather, may lie in the fatigue manipulation. Sensitivity analyses evaluating the influence of varying imputed correlation coefficients showed similar results (Supplemental File 5, Tables S1–S4).

*Hill steepness estimation (n = 3)*
Three experiments (*Bhalla & Proffitt, 1999*; *Proffitt et al., 1995*; *Taylor-Covill & Eves, 2013*) evaluated the effect of fatigue (all running to exhaustion/maximal fitness tests) on steepness estimation of hills (*n* = 2) or stairs (*n* = 1) and were able to be pooled. For verbal measures of steepness (Fig. 3D), participants estimated hills as steeper after fatiguing exercise for both shallow (5°) and steep hills (15–31°). For haptic measures of steepness (Fig. 3E), participants also estimated shallow hills as steeper after fatiguing exercise, but not steep hills. Sensitivity analyses, as described above, showed similar results (See Supplemental File 5, Tables S5 and S6).

*Interim discussion*
There was inconsistent and conflicting evidence for an effect of fatigue on spatial perception of distance, although consistent evidence was seen for hill steepness, with fatigue resulting in reports of increased hill steepness.

For distance spatial perception, pooling found no effect of fatigue, as induced by exercise (Fig. 3A), exercise and sleep deprivation (Fig. 3B) or sleep deprivation alone (Fig. 3C). However, our risk of bias appraisal identified potential issues with fatigue paradigms (*e.g.*, insufficient fatigue for perceptual scaling to occur) and/or distance estimation tasks (*e.g.*, distances not sufficiently challenging for the population). For example, in both pooled studies (*Baati et al., 2020*; *Baati et al., 2015*) that found null results, a relatively easy sprint cycling task (6 s of sprinting × 10), particularly for a young population (mean = 23 years), was used to induce fatigue. In contrast, *Hunt, Hunt & Park (2017)* recruited an older population (mean = 54 years), used a challenging exercise task for that population (stepper task), and used longer distances (92.5 m), and did show findings consistent with EoA hypothesis: fatigued participants estimated distance as further than non-fatigued participants. Exercise task difficulty in context of the population may also have relevance to findings (*Jarraya et al., 2013*) that were *contrary* to what EoA would predict (fatigued athletes estimated distances as being *closer* than when they

were not fatigued). When a highly trained athletic population undertakes a task that is not sufficiently challenging to induce fatigue, it is possible that this increases activation/arousal and perceived energy/vigour (*Thayer, 1978*), inducing an opposite perceptual shift. Given that the studies did not assess whether their fatigue manipulations were successful, it is difficult to know if they were adequate to induce perceptual scaling consistent with the EoA. However, these inconsistent findings may also provide evidence *against* EoA. Indeed, given high risk of bias for participant blinding and given the populations recruited (*e.g.*, often athletes *vs* non-athletes), social effects on compliance with experimental demands and knowledge effects cannot be ruled out. In particular, athletes (such as the soccer players in both *Baati et al. (2015, 2020)* experiments), may have substantial knowledge/ experience in estimating distances. Thus, we cannot rule out the effect of such knowledge on the results.

The "difficulty" of the spatial perception task may also support more consistent findings seen here for hill steepness estimation. Hills are more difficult to walk than flat surfaces (hills requiring more capacity), thus hill estimation may require less 'fatigue' to induce a perceptual shift than flat distance measures. Indeed, pooling of verbal estimations of hill steepness found that people who were fatigued typically estimated hills as being steeper than control groups/conditions (consistent with EoA), although findings were less clear when using haptic measures. Further work is needed to clarify such findings, particularly given that many pooled studies were small, conducted by the same research groups, and the actual fatigue level induced was rarely evaluated.

### *Effect of pain on spatial perception (n = 3)*

*Distance estimation (n = 3)*

Two studies comparing walking distance estimation in people with and without persistent pain (*Tabor et al., 2016*; *Witt et al., 2009*) were pooled. There was no significant effect of pain on distance estimation (Fig. 4). High heterogeneity was seen (*e.g.*, 73%, 83%), where *Witt et al. (2009)* found that people with lower limb pain verbally estimated distances as farther than pain-free controls, and *Tabor et al. (2016)* found no differences between pain and pain-free groups. The third study (*Alaiti et al., 2019*) evaluated the influence of shoulder pain on ratio judgements during a pointing task, where participants estimated the distance to various points on a board (see Table 1 for details). There was no significant effect of shoulder pain on distance estimation (g = −0.33, 95% CI [−0.68 to 0.02]), but a trend towards those with shoulder pain estimating distance as farther than those without shoulder pain was present.

*Interim discussion*

There was limited and inconsistent evidence for an effect of pain on distance perception. The risk of bias assessment identified potential importance of pain by task interactions *i.e.*, that the location of body pain might need to be relevant to the actual performance of the spatial perception task. Participants in *Witt et al. (2009)* had lower limb pain and showed evidence of perceptual scaling for walking distance estimations. In contrast, *Tabor et al. (2016)* recruited people with a variety of chronic pain conditions in various body

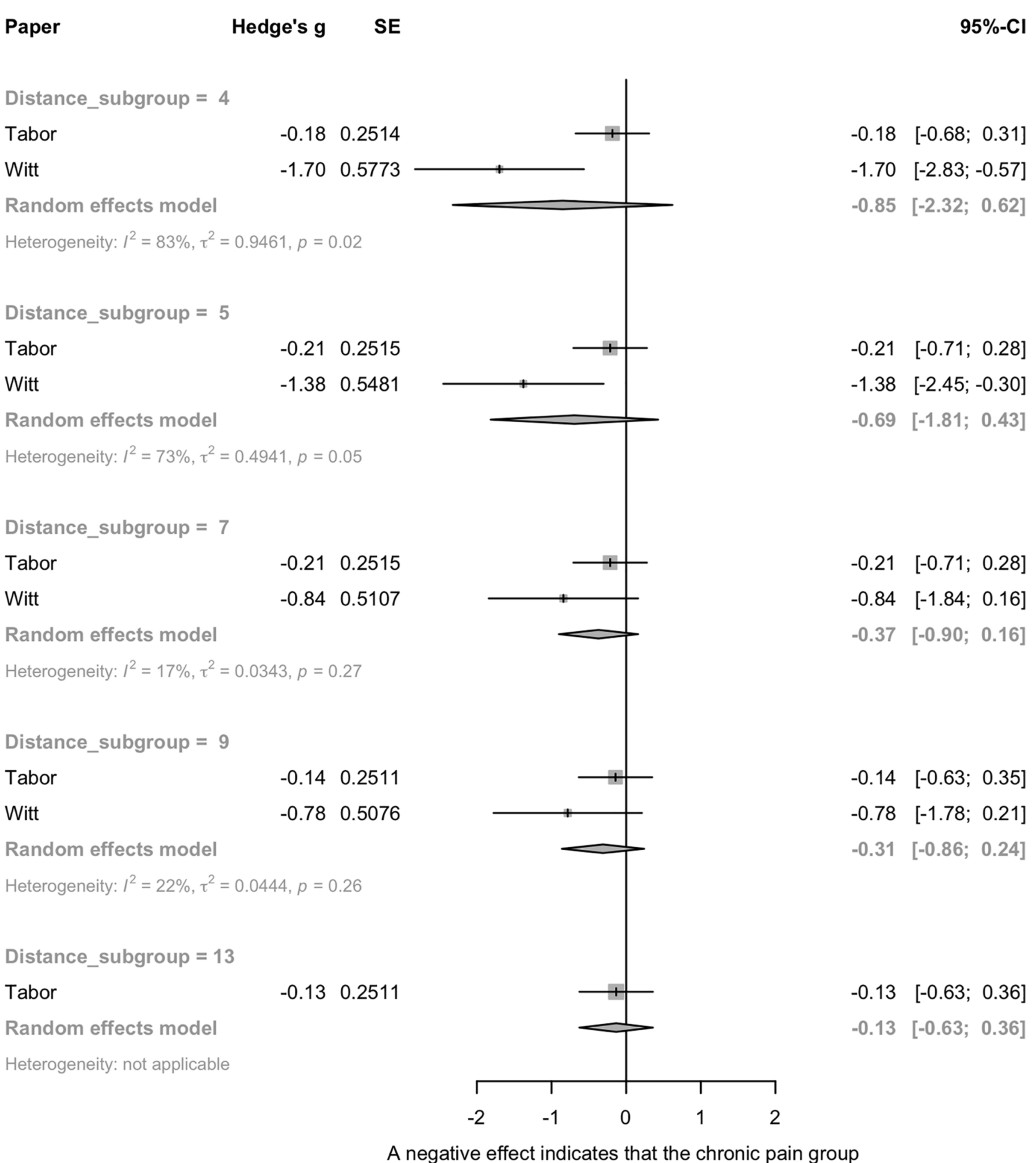

Figure 4 **Pooled estimates for the effect of pain on spatial perception.** A negative effect means that the chronic pain group estimated distances as farther than the control group.

locations, including those whose pain might not interfere with the person's capacity to traverse the environmental distances presented to them, and had null findings. That the spatial perception task might require specificity to the site of pain (*i.e.*, site of reduced bodily capacity), is supported by results found by *Alaiti et al. (2019)* of a trend towards pain-induced effects on a distance perception task that was specific to the area of pain.

Interestingly, a secondary analysis within *Tabor et al. (2016)* showed no distance estimation differences between those that anticipated experiencing pain during walking *vs* those that did not, although it is likely that this *post hoc* analysis was underpowered. Alternatively, pain may be a proxy for another bodily state – *e.g.*, pain-related functional

limitation that results in reduced cardiovascular fitness – that interacts with pain or solely drives changes to spatial perception. Our risk of bias assessment found that none of the studies in this grouping controlled for any confounding variables. Therefore, future studies evaluating numerous putative contributors to spatial perception are warranted.

### Effect of age on spatial perception (n = 10)

*Distance estimations (n = 8)*

Six experiments evaluated verbal estimations of distance and were sufficiently similar to be pooled (*Bian & Andersen, 2013*; *Costello et al., 2015*; *Sugovic & Witt, 2013*). This meta-analysis revealed that older participants estimated distances as farther than younger participants, at all five distances examined (Fig. 5A). There was also a negative effect of age on distance perception at three out of five distances in the single experiment that used an action-based measure (*Bian & Andersen, 2013*); see Fig. 5B. An equidistant cone task was nonsignificant at all distances (*Norman et al., 2020*).

*Hill steepness estimation (n = 2)*

Two experiments (*Bhalla & Proffitt, 1999*; *Dean et al., 2016*) examined the effect of age on measures of hill steepness but were unable to be pooled. They both found a positive relationship when estimating the steepness of moderate hills (*e.g.*, 9 or 10 degrees), indicating that the elderly group estimated the hills as *less steep* than the young group. In an LMM, *Dean et al. (2016)* reported a beta coefficient for age of −0.2°/year, indicating that hill steepness estimates decreased by 0.2° every one year of increased age. However, while *Bhalla & Proffitt (1999)* also found that older participants estimated the hills as *less steep* than younger participants for shallow hills (see Fig. 5C), the opposite was seen for steeper hills (25 or 29 degrees) where older participants estimated hills as *steeper* than younger participants.

*Interim discussion*

While there is consistent evidence that age shifts distance perception in line with the EoA (older participants perceive distances as farther than younger participants), the evidence for hill steepness perception is limited and less consistent. Similar to findings in fatigue studies, a potential task-related influence may be present whereby shallow hills may not be sufficiently "difficult" to induce a challenge to homeostatic state, and thus, show an effect of age, whereas steep hills do show an effect of age. Findings from pooled distance estimations are supportive of an influence of 'difficulty:' older participants over-estimate distance to a greater extent than younger participants (12 m; g = −1.48 (95% CI [−2.04 to −0.91])) and this age-induced difference is smaller for shorter distances (4 m; g = −0.75 (95% CI [−1.19 to −0.30])). However, this does not fully explain findings that older individuals perceive shallow hills as less steep than younger individuals.

Potential limitations exist for pooled distance estimations with four of the six experiments from the same paper (although four different participant groups). While this raises the possibility of systematic error biasing the pooled result, the observed effects in the four experiments were smaller than those found in the others. Thus, if a systematic bias was present, it would be in the direction of a null effect, which provides increased

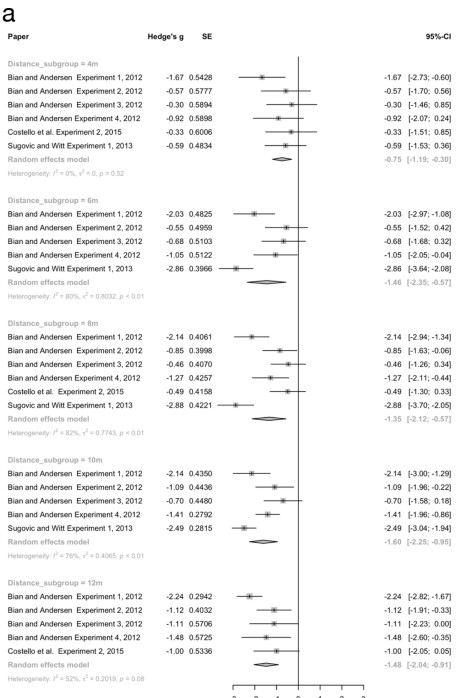

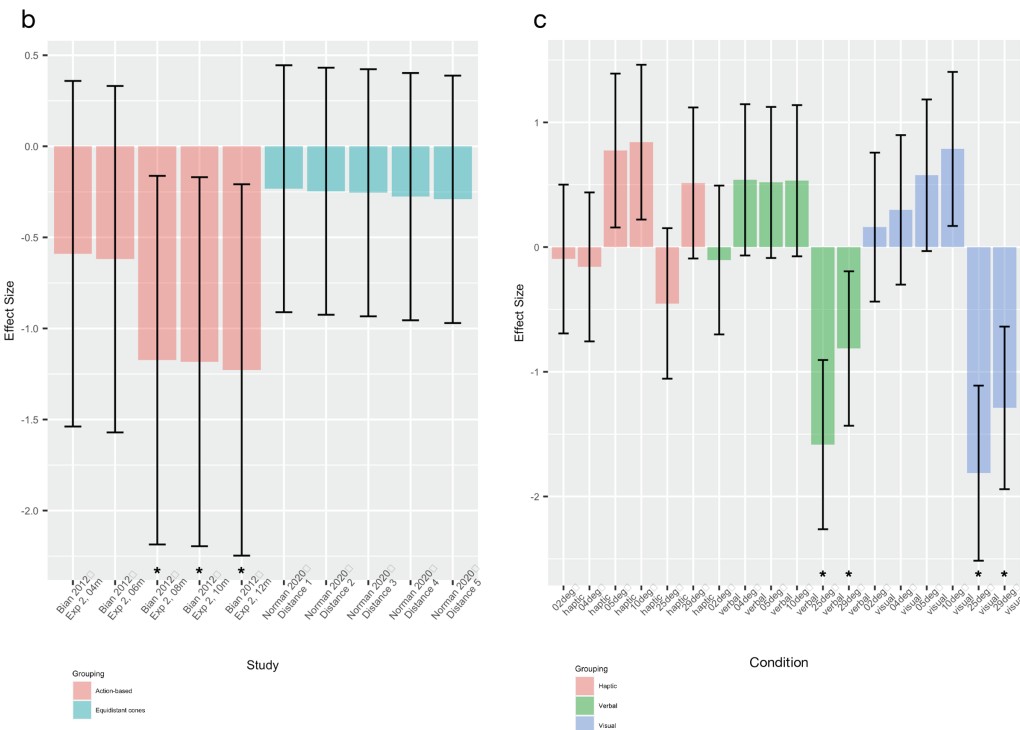

**Figure 5  Pooled estimates for the effect of age on measures of spatial perception.** (A) Verbal distance estimation, subgrouped by distance. (B) Individual study effect sizes. Error bars are 95% CIs and * indicates a significant effect. (C) Effect sizes of all hills from Bhalla (1999), sorted by type of spatial perception measure used. Error bars are 95% CIs and * indicates a significant effect. In all cases, a negative effect indicates that the older group estimated distances as farther than the younger group.

confidence that findings of an effect of age on distance perception are robust. Our risk of bias assessment indicates that potential confounders were poorly controlled in this grouping (all but (*Dean et al., 2016*) scoring high/unclear in this risk of bias domain). Therefore, the observed differences between groups may be due to knowledge effects – where older participants have greater experience in hill and distance estimations than younger participants. Conversely, the between-group study designs employed in this grouping may have reduced response bias, as participants would be less likely to guess the study's hypothesis/anticipated direction of effect.

### Effect of body size/part manipulation on spatial perception (n = 7)

*Distance/aperture width estimations (n = 5)*
Two experiments evaluating the effect of increased hand size (*via* padded gloves) on a visual matching aperture width measure (*Collier, 2018*) were pooled, finding no effect (Fig. 6A). Contrasting results were seen when foot size was manipulated in virtual reality (*Jun et al., 2015*): those in the large foot group estimated gaps as smaller than those in the small foot group. Similarly, across two experiments *van der Hoort, Guterstam & Ehrsson (2011)* used augmented reality (AR) to perform a "body swap" and found that taking on a small body resulted in over-estimation of distance, compared to a normal size body, using both verbal and blind-walking measures. Further, taking on a large body led to underestimation of distance (Fig. 6B). Sensitivity analyses evaluating the influence of varying imputed correlation coefficients showed similar results (See Supplemental File 5, Table S7).

*Hill steepness estimations (n = 2)*
Two experiments from one study (*Bridgeman & Cooke, 2015*) compared hill steepness estimation between conditions of standing on the ground *vs* standing on a 37 cm box. Pooled estimates found that participants standing on a box estimated the hill as less steep than participants standing on the ground, at 2 and 4 m, but not at 8 or 16 m (see Fig. 6C).

*Interim discussion*
While evidence for an effect of body size manipulation on spatial perception was inconsistent, this finding likely reflects high heterogeneity in body manipulation methods, including potential use of inadequate manipulation paradigms. Three of the seven experiments found an effect that supports the EoA at every distance evaluated (and for both verbal and action-based measures), whereby increasing body size decreased perceived distance, and decreasing body size increased perceived distance. Of these, two experiments assessed body size *via* embodiment of AR avatars and confirmed the success of their AR experimental manipulation (participants embodied the large/small bodies).
The remaining four experiments had null or mixed results, although lack of manipulation checks limits the ability to interpret their findings. For example, *Collier (2018)* found that padded gloves (*vs* no gloves) did not change spatial perception. Without a manipulation check, it is unclear whether use of padded gloves solely influenced perceived hand size, whether it resulted in a feeling of encumbrance, or whether it did not alter bodily perception at all. All studies in this grouping scored high or unclear on participant

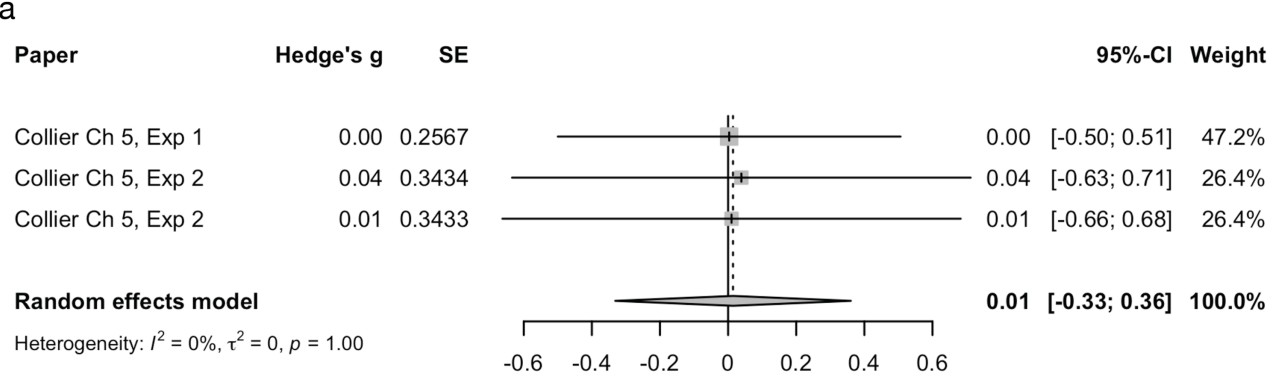

a

| Paper | Hedge's g | SE | | 95%-CI | Weight |
|-------|-----------|-----|---|--------|--------|
| Collier Ch 5, Exp 1 | 0.00 | 0.2567 | | 0.00 [-0.50; 0.51] | 47.2% |
| Collier Ch 5, Exp 2 | 0.04 | 0.3434 | | 0.04 [-0.63; 0.71] | 26.4% |
| Collier Ch 5, Exp 2 | 0.01 | 0.3433 | | 0.01 [-0.66; 0.68] | 26.4% |
| **Random effects model** | | | | **0.01 [-0.33; 0.36]** | **100.0%** |

Heterogeneity: $I^2 = 0\%$, $\tau^2 = 0$, $p = 1.00$

A positive effect indicates that apperatures were estimated as
larger in the unpadded (normal hand) group than the padded (large hand) group

b

Effect Size

Study

Grouping
- Blind-walking
- Verbal

c

| Paper | Hedge's g | SE | | 95%-CI |
|-------|-----------|-----|---|--------|
| **Distance_subgroup = 2m** | | | | |
| Bridgeman and Cooke, Exp 1 | 0.92 | 0.3639 | | 0.92 [0.20; 1.63] |
| Bridgeman and Cooke, Exp 2 | 0.64 | 0.3488 | | 0.64 [-0.05; 1.32] |
| Random effects model | | | | 0.77 [0.28; 1.27] |
| Heterogeneity: $I^2 = 0\%$, $\tau^2 = 0$, $p = 0.58$ | | | | |
| **Distance_subgroup = 4m** | | | | |
| Bridgeman and Cooke, Exp 1 | 0.76 | 0.3581 | | 0.76 [0.06; 1.46] |
| Bridgeman and Cooke, Exp 2 | 0.61 | 0.3479 | | 0.61 [-0.07; 1.29] |
| Random effects model | | | | 0.68 [0.19; 1.17] |
| Heterogeneity: $I^2 = 0\%$, $\tau^2 = 0$, $p = 0.76$ | | | | |
| **Distance_subgroup = 6m** | | | | |
| Bridgeman and Cooke, Exp 1 | 0.47 | 0.3502 | | 0.47 [-0.22; 1.15] |
| Bridgeman and Cooke, Exp 2 | 0.35 | 0.3425 | | 0.35 [-0.32; 1.02] |
| Random effects model | | | | 0.41 [-0.07; 0.89] |
| Heterogeneity: $I^2 = 0\%$, $\tau^2 = 0$, $p = 0.81$ | | | | |
| **Distance_subgroup = 8m** | | | | |
| Bridgeman and Cooke, Exp 1 | 0.19 | 0.3461 | | 0.19 [-0.49; 0.87] |
| Bridgeman and Cooke, Exp 2 | 0.05 | 0.3399 | | 0.05 [-0.62; 0.71] |
| Random effects model | | | | 0.12 [-0.36; 0.59] |
| Heterogeneity: $I^2 = 0\%$, $\tau^2 = 0$, $p = 0.77$ | | | | |

A positive effect indicates that participants standing
on a box percieved the hill as less steep than those standing on the ground

**Figure 6 The effect of body size manipulations on measures of spatial perception.** (A) Pooled estimates for the effect of a hand size manipulation on a visual-matching aperture width task. (B) Individual study effect sizes, sorted by experimental manipulation. Error bars are 95% CIs and * indicates a significant effect. (C) Pooled estimates for the effect of height manipulation on verbal steepness estimations. In all cases, positive effect indicates that the larger body/body part size group estimated distances as closer/hills as less steep than the smaller body/body part size group.

blinding in the risk of bias assessment. Given that most studies in this grouping used within-group designs, lack of blinding may lead to response bias (*i.e.*, participants guessing the hypothesised direction of the effect after being exposed to both manipulation conditions), reducing confidence in these studies' results. Finally, limitations in the experiments manipulating eye height (*via* participants standing on a box) make it difficult to determine whether the results provide evidence against EoA or inconsistent results are due to task constraints. The lack of consistent effect on estimated hill steepness for *all* distances was interpreted as lack of support for EoA (all points on a hill being equally difficult to climb). However, that repeated verbal estimates of the *same, unchanging physical* hill were used, suggests possible response bias contribution.

### Effect of embodiment on spatial perception (n = 3)

*Distance estimations (n = 3)*
Two experiments (*Phillips et al., 2010*; *Ries et al., 2008*) compared distance estimation (*via* blind-walking) of people embodying an avatar (*vs* no avatar) in virtual reality (VR). Pooling found that the avatar group estimated distances as farther than the no-avatar group (Fig. 7A). A third study (*Scandola et al., 2019*) investigated the influence of wheelchair embodiment in people with spinal cord injury (SCI). There was a negative effect of embodiment on distance perception (*via* distance on hill): people estimated distances as farther for steep hills when in their own (embodied) wheelchair *vs* a generic (non-embodied) wheelchair; there was no effect for distances on shallow hills (Fig. 7B).

*Hill steepness estimations (n = 1)*
*Scandola et al. (2019)* also evaluated the effect on hill steepness perception of people with SCI and found no difference between using their own (embodied) wheelchair *vs* a generic (unembodied) wheelchair for visual matching estimation of steepness at any hill angle (Fig. 7C).

*Interim discussion*
Findings were consistent that embodiment (either of an avatar or of a wheelchair) results in perceptual scaling for distance estimates (makes distances seem farther). A strength of these studies was that embodiment paradigms were experimentally evaluated, allowing confidence that the experimental manipulation or condition evaluated the construct it intended to. Only one study (*Scandola et al., 2019*) explored the influence of embodiment on hill steepness perception, therefore replication is needed. Two studies in this grouping (*Phillips et al., 2010*; *Ries et al., 2008*), were conference abstracts and scored poorly on the risk of bias assessment (both scored high or unclear in all categories except for outcome reporting and missing data), thus reducing confidence in the results.

### Effect of glucose manipulations on spatial perception (n = 10)

*Distance estimation (n = 5)*
These experiments were unable to be pooled due to methodological differences, including type of distance estimation measure (action-based, *via* blind-walking and bean-bag toss, and visual matching tasks). Four experiments found that participants in the glucose

a

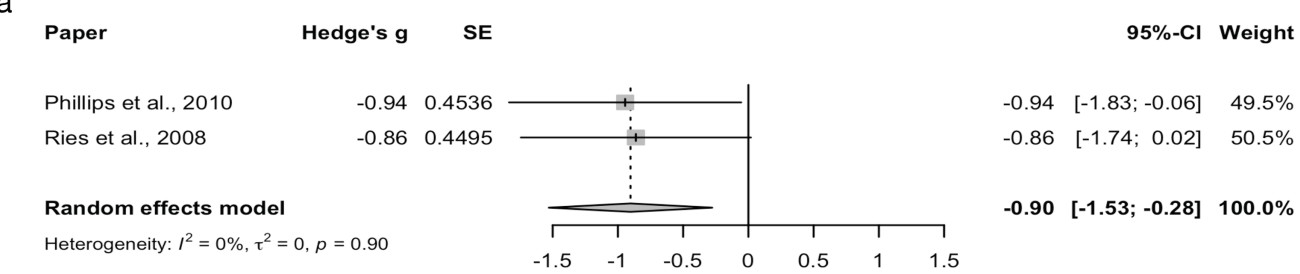

| Paper | Hedge's g | SE | | 95%-CI | Weight |
|---|---|---|---|---|---|
| Phillips et al., 2010 | -0.94 | 0.4536 | | -0.94 [-1.83; -0.06] | 49.5% |
| Ries et al., 2008 | -0.86 | 0.4495 | | -0.86 [-1.74; 0.02] | 50.5% |
| **Random effects model** | | | | **-0.90 [-1.53; -0.28]** | **100.0%** |

Heterogeneity: $I^2 = 0\%$, $\tau^2 = 0$, $p = 0.90$

A positive effect indicates that the avatar group estimated distances as closer than the no-avatar group

b
c

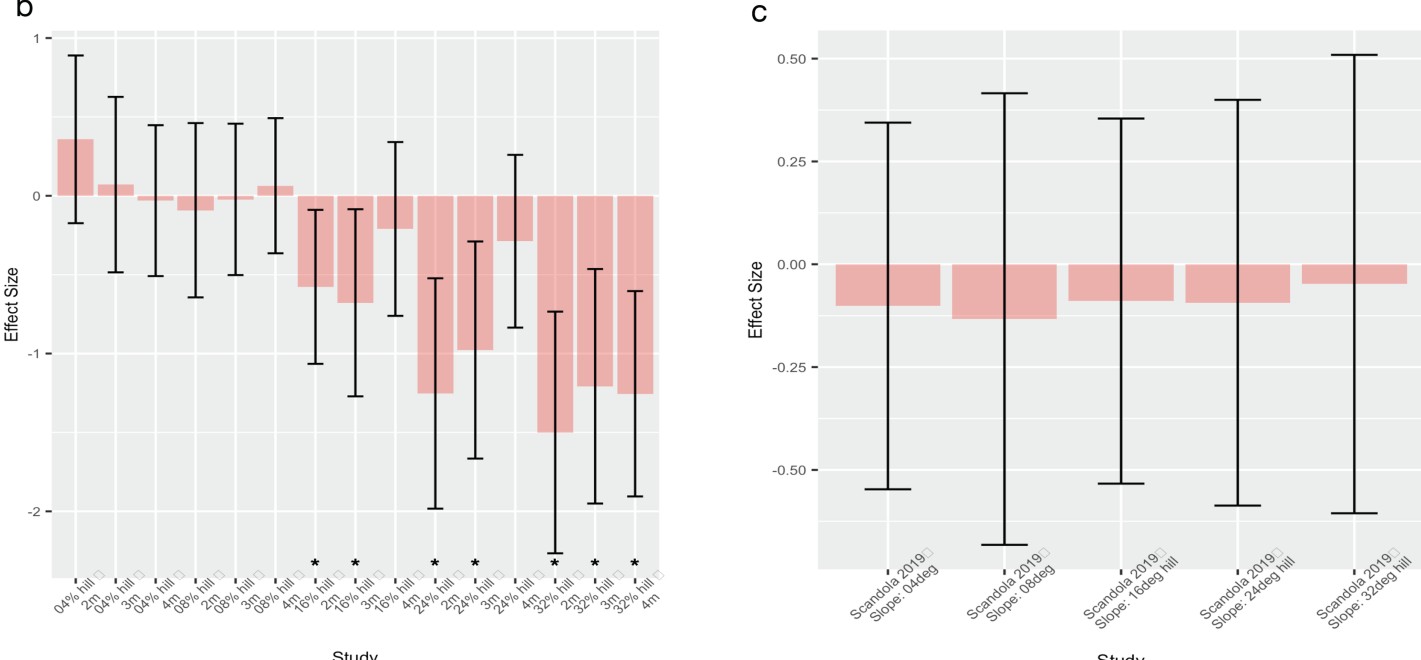

**Figure 7 The effect of embodiment on measures of spatial perception.** (A) Pooled estimates for the effect of embodiment on blind walking distance estimation. (B) Individual effect sizes of embodiment (own *vs* other wheelchair) on verbal estimates of distance on hill estimations. (C) Individual effect sizes of embodiment (own *vs* other wheelchair) on a visual matching hill steepness estimations. Of note, prior to calculating effect sizes from author supplied data for hill steepness estimates, 8.9% ($n = 160$) were removed from the dataset, as the estimations were implausible (>90°, where 90 degrees is a vertical wall). In all cases, a negative effect indicates that the embodied group estimated distances as farther than the not embodied group.

group estimated distances as closer than did those in the control group (placebo glucose; Fig. 8A) (*Cole & Balcetis, 2013*; *Zadra, 2013*; *Zadra, Weltman & Proffitt, 2016*). One experiment had a non-significant result (*Cole & Balcetis, 2013*), but the results were consistent with the overall direction of effect seen in the other studies.

*Hill steepness estimations (n = 5)*
All five studies evaluated verbal hill steepness estimation (*Durgin et al., 2012*; *Schnall, Zadra & Proffitt, 2010*; *Shaffer et al., 2013*; *Zadra, 2013*) and were able to be pooled (Fig. 6A). Ingesting glucose resulted in less steep verbal estimation of hill angle (compared to placebo) for shallow hills, but not steeper hills. There was high heterogeneity in

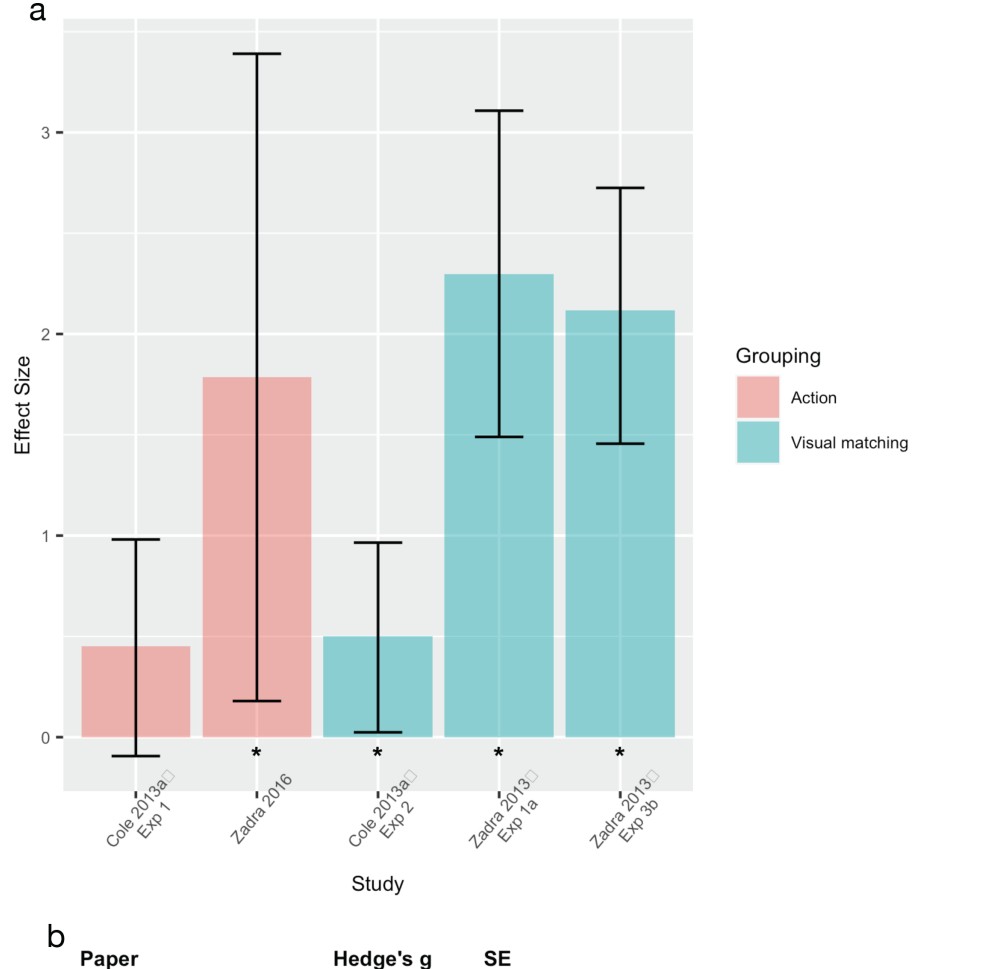

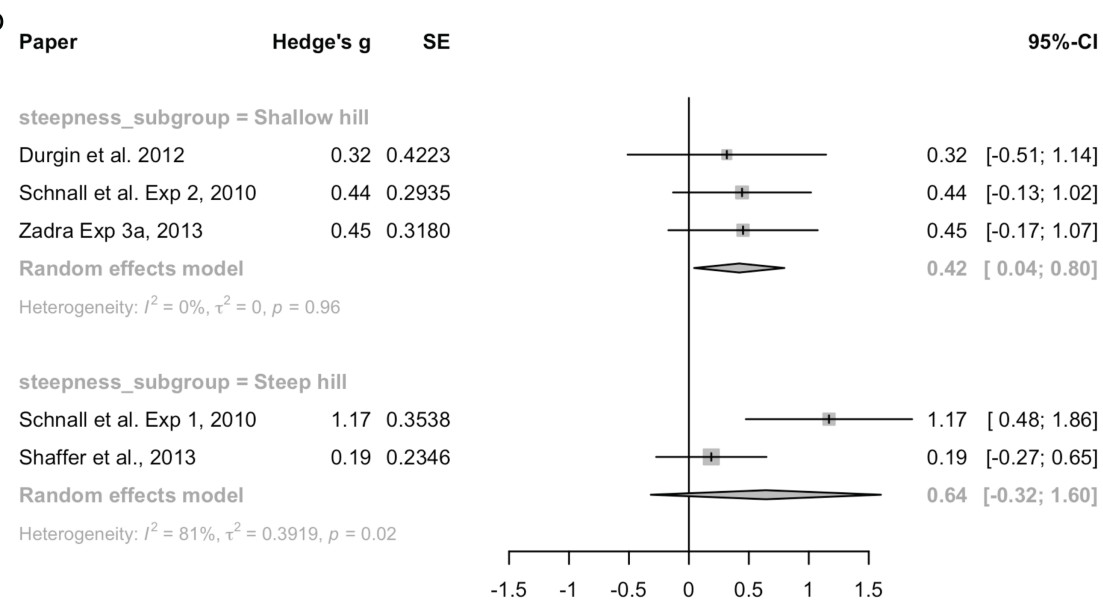

**Figure 8 The effect of glucose manipulations on measures of spatial perception.** (A) Individual study effect sizes, sorted by type of spatial perception measure. Error bars are 95% CIs and * represents significant effects. (B) Pooled verbal estimations of the effect of glucose manipulations on verbal hill steepness, subgrouped by steepness.

pooled analyses of steep hills (I2 = 81%) with one study (*Schnall, Zadra & Proffitt, 2010*) finding a large effect in favour of glucose reducing perceived hill steepness, and one with null results (*Shaffer et al., 2013*).

*Interim discussion*

There was consistent evidence that participants who ingested glucose estimated distances as shorter and hills as less steep than in control groups/conditions – in line with the EoA. While a non-significant pooled effect was found for the steep hill subgroup, the study that found a small non-significant effect (*Shaffer et al., 2013*) was assessed to have a *higher* risk of bias (did not control for confounders, and missing data >15%) than the study that did find a significant effect of glucose (*Schnall, Zadra & Proffitt, 2010*), thus more weight is placed on the latter's findings. Furthermore, in all but one study (*Cole & Balcetis, 2013*), participants ingested food/drink either containing glucose or a sugar free sweetener (placebo) allowing participant blinding. In addition, this work contained the only two experiments (*Cole & Balcetis, 2013*; *Zadra, 2013*) where assessors were blinded. Adequate blinding and the use of between-group study designs means that competing theories (experimental demands and response bias) are less likely to have influenced the results, increasing confidence that the findings do reflect an effect of glucose on measures of spatial perception.

### Fitness and measures of spatial perception (n = 11)

Experiments used either objective proxies of fitness (*e.g.*, BMI or waist-to-hip ratio), or perceived fitness (*e.g.*, subjective ratings of fitness or body size). Given the diversity of measurements, we standardised the direction of effects as follows: a negative effect would indicate that *increased* actual/perceived fitness resulted in *decreased* measures of spatial perception (distances shorter, hills less steep).

*Distance estimation (n = 4)*

Methodological differences prevented pooling. All effects sizes were negative, but only two significant (Fig. 9A). One study (*Sugovic, Turk & Witt, 2016*) evaluated the effect of BMI on verbal distance estimation finding significant effects for estimates at 10 m, but not at any other, longer distances, as well as no effect of perceived body size (picture selection task) at any distance. *Taylor, Witt & Sugovic (2011)* found that traceurs/parkour athletes (*i.e.*, higher fitness/sport-specific training) estimated vertical wall height as significantly lower than untrained controls for a 1.94 m wall but not for higher walls (2.29 m, 3.45 m). Effect sizes were unable to be calculated for two other studies, but one found a non-significant relationship between waist-to-hip ratio and distance perception (*Cole, Balcetis & Zhang, 2013*), and one found no relationship between self-rated fitness and a distance on hill measure (*Shaffer, Greer & Schaffer, 2019*).

*Hill steepness estimations (n = 7)*

Three experiments (all from the same paper) that evaluated the effect of self-rated fitness on verbal (Fig. 9B) and haptic (Fig. 9C) measures of hill steepness were able to be pooled (*Krpan & Schnall, 2014*) and produced non-significant results. The verbal measure

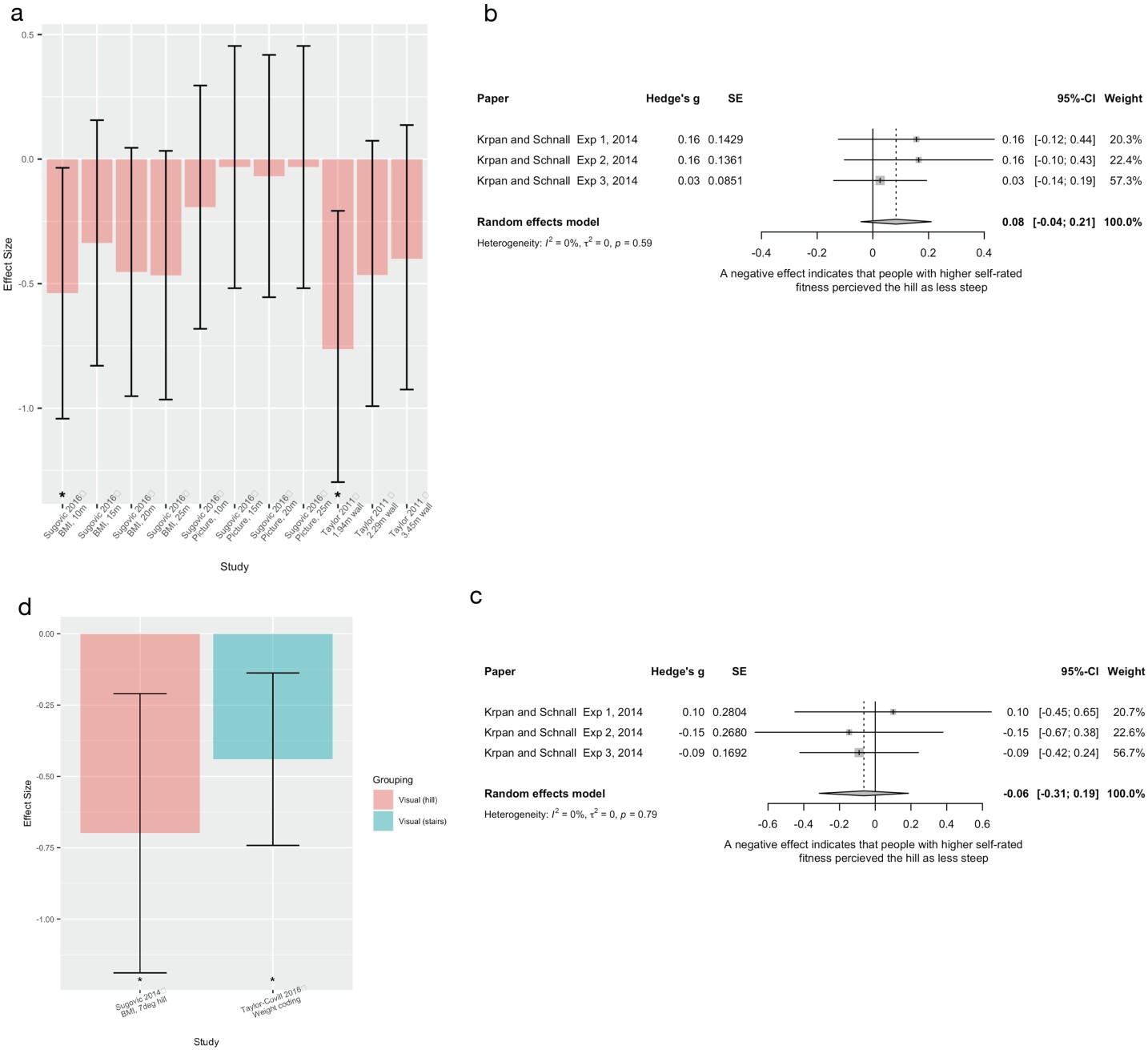

**Figure 9 The effect of fitness loads on measures of spatial perception.** (A) Individual distance estimation experiment effect sizes. Error bars are 95% CIs and an asterisk (*) indicates significant effects. (B) Pooled estimates for self rated fitness and verbal hill estimation. (C) Pooled estimates for self-rated fitness and haptic hill estimations. (D) Individual steepness estimation experiment effect sizes. Error bars are 95% CIs and an asterisk (*) indicates significant effects. In all cases a positive effect indicates that the higher fitness group estimated measures of spatial perception (distances and steepness) as larger than the lower fitness group.                

trended towards a positive effect, indicating that people who rated their own fitness as higher, also perceived the hill as steeper; while the haptic measures trended in the opposite direction, towards a negative effect.

The remaining studies used objectively measured weight/fitness and found effects on perceived hill steepness (Fig. 9D). *Sugovic (2013)* found that hills were estimated as steeper as BMI increased. Similarly, *Bhalla & Proffitt (1999)* combined BMI and a submaximal fitness test to create a composite "fitness measure", and found that as fitness increased, estimated hill steepness (verbal) decreased. Haptic responses had no relationship with fitness measurements. In another experiment (*Taylor-Covill & Eves, 2016*) experimenters used weight coding to group participants into "clearly overweight" and "healthy" groups, and also found that overweight status increased perceived hill steepness estimates. Likewise, *Dean et al. (2016)* found that people with higher BMI gave higher slant estimations (LMM beta = 0.5). A final experiment (*Taylor-Covill & Eves, 2016*) also found an association (*via* LMM) between body composition (measured *via* dual-energy x-ray absorptiometry [DEXA] at baseline and follow-up; average of 407 days between assessments) and perceived steepness of staircases. At both timepoints BMI and fat mass were positively associated with verbal and visual measures (increases in BMI/fat mass associated with increased estimated steepness), but the model was non-significant for haptic measures of steepness. Longitudinally, when BMI and body fat decreased, so too did verbal estimations of steepness, but not haptic.

*Interim discussion*

There was no evidence for a relationship between *self-rated* fitness and spatial perception, but there was consistent evidence for associations with objective fitness measures.

Limitations in the measurement of self-rated fitness likely contribute to the lack of effect. For example, *Krpan & Schnall (2014)* assessed perceived fitness (self-rated) using a 5-point scale, and had poor distribution of responses (*e.g.*, in experiment 1, 82.7% of participants rated their condition as either good [4] or excellent [5]), and thus, likely has poor sensitivity to differences in perceived fitness. The perceived fitness outcome measures used by *Sugovic, Turk & Witt (2016)* were similar, with poorly defined constructs (*e.g.*, body type category of Large/Muscular) and limited response options. Unclear validity of the fitness measure cannot be ruled out as a contributor to the lack of relationship between fitness and distance measures.

While more consistent evidence in line with EoA was present for objective measures of fitness, such as BMI, waist-to-hip ratio, and fat mass influencing spatial perception, methods of assessing objective fitness were also problematic. Authors reported measuring "physical abilities" *via* BMI (*Sugovic, Turk & Witt, 2016*) and "physiological fitness" *via* waist-to-hip ratio (*Cole & Balcetis, 2013*). While these measures hold correlations with measures of fitness (*Wier et al., 2006*), they have important limitations that may be relevant to EoA. For example, BMI does not account for differences between muscle and fat. Thus, a heavily muscled, but fit individual (*e.g.*, Lebron James, BMI 27.5; overweight category), would have a higher BMI then a sedentary individual, with low muscle mass and high body fat, but we would expect very different results for spatial perception based on the EoA hypothesis (*Nevill et al., 2006*). One study evaluated "body composition" (*Taylor-Covill & Eves, 2016*) using DEXA scanning, which does allow for the delineation between muscle and fat. Indeed, this study found that fat mass, rather than fat free mass, was

significantly associated with measures of spatial perception both cross-sectionally and longitudinally. Additionally, consistent with other groupings, these studies scored high risk of bias for the blinding domain. Thus, it is difficult to rule out the influence of response bias in these results.

### External load and measures of spatial perception (n = 12)

Seven studies used backpacks (all approximately 20% of participants body weight), four used ankle weights (5–10% of body weight), and one used resistance bands.

*Distance and gap estimation (n = 11)*
Pooling found that external loads resulted in participants estimating distances as farther than no load groups/conditions when using verbal (Fig. 10A) and VR based walking distance matching (Fig. 10B) measures, but not blind walking measures (Fig. 10C). An additional study evaluated within-group differences in verbal estimates of distance when wearing a backpack (*Vinson et al., 2017*), with findings consistent with pooled results (unable to calculate effect size). The remaining experiments were also unable to be pooled (Fig. 10D). Of these, two (*Corlett, Byblow & Taylor, 1990*; *White, 2012*) compared medium load (5 kg ankle weight/moderate resistance band) conditions to high load (10 km ankle weight/heavy resistance band) conditions. Both experiments had small, non-significant effects in the direction that the high load group perceived distances as farther during blind walking measures (*Corlett, Byblow & Taylor, 1990*) and VR based distance matching (*White, 2012*). Last, one experiment found that participants verbally estimated gap widths to be wider while wearing ankle weights than they did without (*Lessard, Linkenauger & Proffitt, 2009*). Sensitivity analyses evaluating the influence of varying imputed correlation coefficients showed similar results (Supplemental File 5, Tables S8 and S9).

*Hill steepness estimations (n = 1)*
A single study evaluated the effect of external loads on verbal estimation of hill steepness (*Bhalla & Proffitt, 1999*), finding significant effects for a 5° hill but not a 31° hill (Fig. 10D). Opposite effects were seen for haptic measures of hill steepness.

*Interim discussion*
There was consistent evidence that external loads (*vs* no load) influence verbal and VR-based measures of spatial perception in line with EoA, but conflicting evidence for studies evaluating varying degrees of loading. Results require cautious interpretation given experimental demands and response bias. When two experiments (*Keric & Sebanz, 2021*) used identical protocols, but one used a cover story (successfully blinding participants) and one did not (all participants guessed study hypothesis), the effect estimate was heightened in favour of the unblinded group in line with EoA predictions (g = −0.32 *vs* −0.01 in the blinded group). According to our risk of bias assessment, participants were not blinded in the remaining studies in this grouping. Given that these unblinded studies have similar, large negative effects to the unblinded *Keric & Sebanz (2021)* experiment, confidence in the overall pooled effect is reduced. Further use of

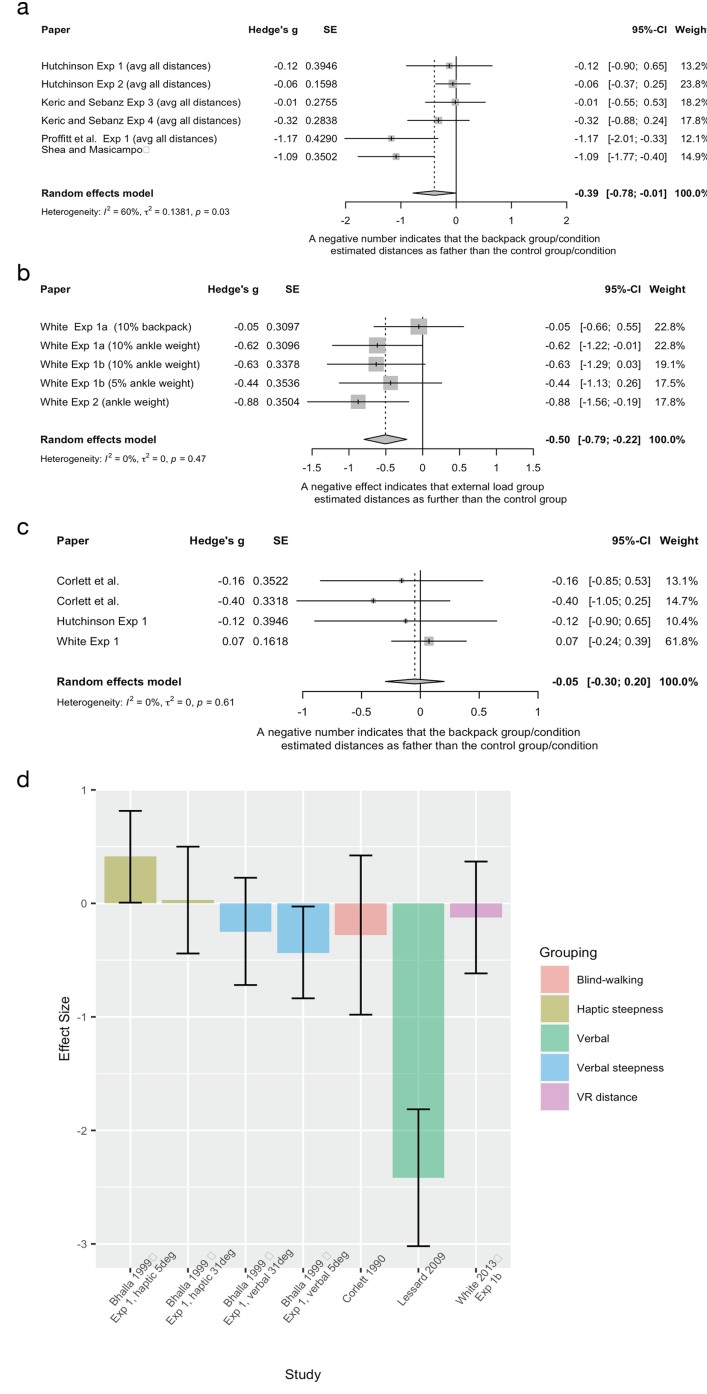

**Figure 10** **The effect of external loads on measures of spatial perception.** (A) Pooled estimates of the effect of external loads on verbal estimation of distance. (B) Pooled estimates of the effect of external loads on a VR matching distance paradigm. (C) Pooled estimates of the effect of external loads on blind0-walking distance estimation. (D) Individual study effect sizes. Error bars are 95% CIs and an asterisk (*) indicates significant effects. In all cases a negative effect indicates that the external load group estimated measures of spatial perception (distances and steepness) as larger than the control group.

within-group, repeated measure design of some studies (*e.g.*, (*Lessard, Linkenauger & Proffitt, 2009*)) increase the possibility of response bias driving the effects seen.

Pooled results for VR experiments manipulating external load also showed a significant effect in line with EoA predictions; however, all studies were from the same author (*i.e.*, four comparisons from two experiments), and are from a dissertation (not peer reviewed). Given sample size reductions necessary for pooling, this resulted in small samples ($n = 9$–$17$), increasing the risk of spurious results (*Higgins et al., 2021*). Additionally, assessor participation in measures of spatial perception (*e.g.*, assessor stopped the treadmill during VR-based walking distance matching), paired with assessor non-blinding to test condition/study aim heightens the possibility of measurement bias in this method.

Studies (*Corlett, Byblow & Taylor, 1990*; *White, 2012*) investigating varying degrees of external loads (high, low, no load) had non-significant results, but failure to assess participant-specific perceptions of load magnitude hampers clear conclusions. It is unclear whether trends towards increased distance estimation with high *vs* low loads represent the possibility of a dose-response relationship, perhaps uncaptured if the loads applied were too light to induce a homeostatic threat/influence (*e.g.*, standard high load may not be perceived as high/difficult for some people).

### *Effect of interoception (n = 2)*

Two studies evaluated perceived internal bodily state (*i.e.*, interoceptive accuracy) assessed using a heart rate accuracy test.

*Distance estimations (n = 1)*

The high interoceptive accuracy group (two-thirds of participants) perceived the same distance as significantly farther on a hill than on the flat, in line with the EoA (hills require higher energetic cost, therefore the same distance should be perceived as further than on flat), but the low interoceptive group (remaining one-third of participants) did not (*Tenhundfeld & Witt, 2015*). Between-group differences were significant at only one of the four distances (Hedge's g at 2 m = 0.25 (95% CI [− 0.35 to 0.85]), 4 m = −0.52 (95% CI −1.12 to 0.09]), 6 m = −0.64 (95% CI [−1.25 to −0.03]), and 8 m = −0.6 (95% CI [−1.21 to 0.01])).

*Hill steepness estimations (n = 1)*

A significant association was found between interoceptive accuracy and verbal estimations of hill steepness (g = −0.89, 95% CI [−1.36 to −0.41]), indicating that those with higher interoceptive accuracy had lower hill steepness estimations (over-estimated steepness to a lesser degree) (Mouatt et al., 2020, unpublished data).

*Interim discussion*

Very limited evidence exists that an individual's ability to perceive their own internal bodily state influences spatial perception. One study (*Tenhundfeld & Witt, 2015*) had high risk of bias (all domains assessed as high/unclear), and during correspondence, it was reported that a replication study failed to demonstrate an effect (N. Tenhundfeld, 2021, personal communication). Thus, results here should be interpreted with caution.

### Assessment of publication bias

Contour enhanced funnel plots for each forest plot are given in Supplemental File 6. Only the funnel plot for age met the suggested requirements of a minimum of 10 data points to allow valid interpretation (*Higgins et al., 2021*). While funnel plot is symmetrical around the summary effect size, several points are outside the $p < 0.01$ boundary, which may indicate publication bias. Due to significant concerns of publication bias in the field (*Schäfer & Schwarz, 2019*), we created an additional post-hoc funnel plot including all studies with available data. In studies with multiple comparisons, we chose the comparison that was most consistent with the overall result (if the majority comparisons were non-significant, we chose a non-significant effect estimate; if the majority comparisons were significant, we chose a significant effect estimate; if there was a mix of results, we chose an effect estimate at random). Direction of effect was standardised such that a negative value indicates that a study's result supports the EoA theory. One study was a clear outlier (effect size = 9.91); the funnel plot was completed both with and without this study, with the latter allowing more robust visual inspection of symmetry. Overall, there was adequate symmetry. However, 17% ($n = 9$) of points were outside the $p < 0.01$ boundary in the direction of a negative effect, which may indicate publication bias favouring the EoA hypothesis.

## DISCUSSION

We found mixed support for the EoA hypothesis – that the environment would be perceived as harsher (distances further, hills steeper) with reduced body capacity – across various bodily states. The strongest and most consistent evidence in favour of the EoA was found for age and glucose manipulation. There was mixed evidence for embodiment, body size manipulations, fitness, external loads, and interoception. Our risk of bias assessment indicates that theories of social effects could only be ruled out in studies evaluating the effect of glucose due to their use of adequate blinding procedures. There was no support for fatigue influencing spatial perception measures, given highly conflicting evidence. However, lack of confirmation of successful induction of bodily states such as fatigue, limit conclusions. That is, it remains unclear whether the tested bodily state does not influence spatial perception or whether the tested bodily state was not actually successfully manipulated. Moreover, given high risk of bias for all included studies, particularly in relation to response bias and use of heterogenous spatial perception measures with unknown reliability, the existence of an effect of any bodily state on spatial perception remains uncertain.

Our meta-analytical results are consistent with those of a previous review of more narrow focus (*Molto et al., 2020*), which found evidence to support a small effect on visuospatial perception of action-constraint (one's ability to act within the environment). This previous review also investigated two commonly cited rival hypotheses, which both posit that the observed effects for visuospatial perception are due to experimental demand biases, rather than reflecting perceptual changes. The first hypothesis suggests that participants are more likely to correctly guess the hypothesis in within-group designs than between-group designs, and the second that verbal measures of spatial perception

may be more sensitive to voluntary control than other measures (*e.g.*, visual-matching or action-based). Thus, it is hypothesised that the observed effects on spatial perception should be larger in studies using within-group designs and verbal measures. Their moderator analysis did not support either hypothesis (*Molto et al., 2020*). However, based on our findings of high overall risk of methodological bias of these studies, we argue that such comparisons may be premature. That is, the null results of their moderator analyses may reflect high variability of effects due to high risk of bias, rather than a true null result.

Despite methodological issues of included studies, the current review revealed important implications for the field of perceptual research. Across various bodily states, we found evidence that perceptual scaling may require an interaction between bodily state and environmental "difficulty". That is, perceptual scaling in line with the EoA may require both reduced bodily capacity and an environment that is sufficiently "harsh" to induce homeostatic threat. For instance, in studies evaluating fatigue, scaling occurred in hill steepness estimations but not in distance estimations; this may represent insufficient "difficulty" of flat surfaces to evoke a threat to homeostasis needed to scale visual perception when participants are likely not overly fatigued. The results for age-induced effects on distance estimations also support this idea, where effect sizes increased as the target distance (and thus homeostatic threat) increased. Homeostatic specificity, and the interaction between bodily state and perceptual task, was also seen in studies evaluating pain, where the results suggest that the measure of spatial perception must be relevant to the site of pain to capture perceptual scaling (*e.g.*, lower limb pain and walking distance estimation, or shoulder pain and pointing distance estimation). Future work should ensure that the environmental stimuli is both appropriately "difficult" and homeostatically relevant to induce scaling given the target population. Exploring thresholds for difficulty and homeostatic relevance on perception also merits consideration.

The interaction between bodily capacity and environmental features is also seen in cases where perceptual scaling did *not* occur. In a study of Parkour athletes (*Taylor, Witt & Sugovic, 2011*), estimations of wall heights were compared to an untrained control group, with a significant perceptual benefit seen for athletes only for the shortest wall (1.94 m) but not higher walls (2.29 m, 3.4 m). At this low height, there may be a differential ability between groups to climb the wall, enabling perceptual scaling, whereas for the higher walls, climbing was equally unachievable for both groups, and perceptual scaling was absent. Similar results occurred for distance estimations of gap width in weighted *versus* unweighted conditions (*Lessard, Linkenauger & Proffitt, 2009*). A significant difference in distance estimation occurred, favouring smaller gap width estimation for the unweighted condition, but disappeared when gaps were *outside* of participants' estimated jumping ability (too far to jump across).

That similar bodily manipulations did not have uniform effects on perception, but rather, were specific to the perceptual task provides support for EoA. For example, past work has shown that embodying a small avatar (or the avatar of a child), increases estimated size of objects (*Banakou, Groten & Slater, 2013*). Indeed, the findings of this review, where embodying a small avatar increased perceived distance, extends this work,

and suggests a similar perceptual scaling mechanism, where larger bodies signify greater bodily capacity (*van der Hoort, Guterstam & Ehrsson, 2011*). That we view the world in context of our body is also supported by review findings for embodiment showing that *having* a body in virtual space is a requirement for spatial scaling to occur (*Phillips et al., 2010*; *Ries et al., 2008*; *Scandola et al., 2019*). Without a body, there is feasibly no internal model that can inform capacity, and thus no alteration in environmental perception. If this latter aspect is true, there are clear implications not only for the study of perception, but also in the development of virtual reality applications.

Our findings also have important methodological implications for the broader field of embodied perception. First, future research should ensure that experimental manipulations of bodily state are achieved as intended. For example, when evaluating body size, assessing induced alterations to perceived body size is needed to ensure successful manipulation. Additionally, consideration and control of potential confounders is needed. The results of this meta-analysis indicate that many factors relating to bodily state may influence spatial perception, and therefore may act as confounders. For instance, if one was interested in the influence of pain on spatial perception, ensuring that the groups are age-matched to control for the confounding influence of age on spatial perception is warranted. Or, when assessing the influence of fatigue on spatial perception, it may also be relevant to consider body size, glucose levels, and/or fitness levels, as these could influence perceptual scaling directly or *via* interactions with fatigue. Other factors, such as affective state, may also be relevant to control for, however, it is beyond the scope of this review to make such recommendations. Regardless, *a priori* consideration and pre-registration of these variables is important for sample size calculations and to avoid type 1 error. Interestingly, use of augmented or virtual reality to manipulate bodily state often significantly influenced perception (*Jun et al., 2015*), while similar real-life manipulations did not (*Collier, 2018*). Determining if such conflicting results represent virtual environments not translating to real-life, or insufficient/unbelievable real-life paradigms used is warranted.

Second, one consequence of the criticisms around measurement of spatial perception (*i.e.*, that 'judgements' reflect response bias rather than perceptual effects, as previously highlighted) is that indirect measures of spatial perception, such as action-based measures (blind walking, throwing tasks) or visual matching paradigms, are favoured because they are thought more robust against response bias. A presumed strength of these tasks is that participants use novel motor skills (walking with a blindfold or throwing a bean bag), with task novelty reducing the ability of participants to cognitively conform to perceived experimental demands. However, such novelty has limitations. Motor learning, or the process through which one develops the ability to perform a new action, is complex; accurately and consistently performing a motor task requires time and practice (*Krakauer et al., 2019*). Thus, observed effects for these measures may reflect poor and unreliable performances of the action-based task itself, rather than perceptual re-scaling. For example, even if an individual perceived a distance to be near, a beanbag toss as the measure of spatial perception ultimately relies upon their ability to avoid "overshooting" or

"undershooting" the target. Whether action-based tasks provide an accurate representation of perceptual shifts is yet to be elucidated.

Challenges with reliability extends beyond action-based tasks. Our review found that only four studies used spatial perception measures that had previously established reliability (See Table 2). While the call for reliable tasks is not new – it was first discussed in 2013 and in many subsequent papers (*Firestone, 2013*; *Philbeck & Witt, 2015*; *Witt, 2020*) – the continued lack of reliability assessment for spatial perception measures means that results are difficult to interpret. For example, it is uncertain whether null results relate to a lack of precision of the spatial measure or lack of perceptual effects. Ensuring reliability of spatial perception measures is a priority for future research. Given our findings, we strongly recommend that expert consensus, using methods such as a Delphi survey, to establish guidelines around methodological considerations for reducing response bias and measuring spatial perception of the environment is essential to move this field forward. Ideally, guideline development would involve experts from many fields, including motor learning and blinding, to ensure that proposed strategies have methodological rigour.

This review has several methodological strengths. Our protocol was developed *a priori* and uploaded to Open Science Framework, with any deviations to the protocol disclosed and reported. We used best practice methodology *via* compliance to the PRISMA 2021 statement (*Page et al., 2021*). Two independent reviewers completed study screening, risk of bias assessment, and data extraction. To minimise bias in the meta-analyses, comparisons chosen and groupings for pooling were determined by an assessor blinded to the results of individual experiments. The search strategy was comprehensive, and we contacted authors from 36 of 45 included studies to obtain missing data necessary for quantitative analysis.

Limitations of this review include the possibility of missing relevant studies given the lack of established database subject headings for bodily state. However, this risk was mitigated using a thorough systematic search strategy, informed by an academic librarian, including pearling of reference lists, as well as consideration of grey literature. As with all meta-analyses, there is risk that effect sizes may be artificially inflated as this method can amplify publication bias (*Friese & Frankenbach, 2020*). Within psychology, there is a general consensus that publication bias is present, however the exact prevalence is unknown (*Schäfer & Schwarz, 2019*). Therefore, it is difficult to estimate the degree to which these factors influenced the results of this study. We attempted to mitigate the effects of publication bias by including grey literature, however there may still be systemic error present in our results. Additionally, a possible limitation exists *via* the graphical extraction of outcome data using electronic tools in experiments where numeric data were not available. However, our chosen method has good reliability (*Drevon, Fursa & Malcolm, 2017*), and any errors occurring from extraction are unlikely to be systematic. Last, it is possible that our effect estimates for within-group studies were influenced by imputation of the correlation between the pair of observations ($r$). We were able to calculate the value of $r$ from raw data in 23 comparisons, but for the remaining we imputed $r$ using values from similar experiments, as recommended (*Higgins et al., 2021*).

Importantly, that our sensitivity analyses (increasing and decreasing *r* by 0.1) did not change our results, suggests that our findings are robust.

## CONCLUSION

This meta-analytical systematic review found low-level evidence for the effect of bodily state on measures of spatial perception, given the high risk of bias of eligible studies. However, consistent evidence exists to support that glucose manipulations and age influence spatial perception of distance and hill steepness in line with the Economy of Action hypothesis where lower capacity results in over-estimation of distance/steepness (and *vice versa*). In the glucose grouping, adequate blinding and study design reduces the likelihood of competing theories (social and knowledge effects) explaining the results, thus increasing the confidence that the results reflect true shifts in spatial perception. Methodological limitations of this field, including the lack of confirmed success of bodily state manipulations and the lack of consideration of relevant confounders, make interpretation of both significant and null results challenging. Establishing the reliability of spatial perception measures and employing experimental methods to reduce participant response bias (including blinding) are priorities for future research.

### Funding

Erin MacIntyre is supported by the University of South Australia Post Graduate Award (USAPA) and by a National Health & Medical Research Council Project Grant to Tasha R. Stanton (ID 1161634). Brendan Mouatt is supported by a Leadership Investigator Grant from the National Health & Medical Research Council of Australia to GLM (ID 1178444). Felicity A. Braithwaite is supported by the John Stuart Colville Fellowship (Arthritis Foundation of South Australia). Tasha R. Stanton is supported by the National Health & Medical Research Council (NHMRC) of Australia Career Development Fellowship (ID1141735). The funders had no role in study design, data collection and analysis, decision to publish, or preparation of the manuscript.

### Grant Disclosures

The following grant information was disclosed by the authors:
University of South Australia Post Graduate Award.
National Health & Medical Research Council Project: 1161634.
National Health & Medical Research Council of Australia to GLM: 1178444.
John Stuart Colville Fellowship (Arthritis South Australia).
National Health & Medical Research Council (NHMRC) of Australia Career Development Fellowship: 1141735.

### Competing Interests

Felicity A. Braithwaite and Tasha R. Stanton have received speaker fees for providing lectures related to pain and rehabilitation. Brendan Mouatt receives renumeration from

The Knowledge Exchange for provision of continuing professional development workshops related to pain and exercise rehabilitation. All other authors report no conflicts of interest.

## Author Contributions

- Erin MacIntyre conceived and designed the experiments, performed the experiments, analyzed the data, prepared figures and/or tables, authored or reviewed drafts of the paper, and approved the final draft.
- Felicity A Braithwaite conceived and designed the experiments, performed the experiments, analyzed the data, prepared figures and/or tables, authored or reviewed drafts of the paper, and approved the final draft.
- Brendan Mouatt performed the experiments, analyzed the data, prepared figures and/or tables, authored or reviewed drafts of the paper, and approved the final draft.
- Dianne Wilson performed the experiments, analyzed the data, prepared figures and/or tables, authored or reviewed drafts of the paper, and approved the final draft.
- Tasha R. Stanton conceived and designed the experiments, analyzed the data, authored or reviewed drafts of the paper, and approved the final draft.

## Data Availability

The protocol, amendments to protocol, search tracking and full text screening, blank risk of bias and data extraction forms, data and R code for meta-analysis are all available at OSF:

MacIntyre, E., Mouatt, B., Stanton, T., Braithwaite, F. A., & Wilson, D. (2022, February 28). The influence of real and perceived bodily state on spatial perception of the external environment: a systematic review. Retrieved from https://osf.io/6zya5/.

## Supplemental Information

Supplemental information for this article can be found online at http://dx.doi.org/10.7717/peerj.13383#supplemental-information.

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
