# Peer review of "Does who I am and what I feel determine what I see (or say)? A meta-analytic systematic review exploring the influence of real and perceived bodily state on spatial perception of the external environment"

_PeerJ, doi:10.7717/peerj.13383_

## Round 0.1 · original submission · Major Revisions

I have received thoughtful reviews from three experts in the field. Frank Durgin is Reviewer 2. I thank the reviewers for the time and energy they devoted to sharing their insights on the piece. The reviewers all express some concerns about the manuscript. However, I believe that the concerns can be addressed in a revision that would allow the manuscript to reach threshold for publication at PeerJ.

I will synthesize some of the critiques from reviewers here, but the full reviews appear below.

The reviewers and I believe you should include more discussion that situates findings within a theoretical context, including the myriad exemplars of embodied theories (see Reviewer 3) and the theories that conflict with these models (see Reviewer 2).

I applaud your work to pre-register your analysis. However, there are places where the pre-registration was under-specified. Please clearly note which decisions moved beyond the pre-registration, like the ad hoc groupings that were established. Further, please include a clearer articulation of what is meant by “bias” at each point when the term is used.

Finally, it is important to access the grey literature to counteract the file-drawer problem, but there is value in being clearer about which findings were peer-reviewed and which weren’t, given that the synthesis is happening by subgroup.

Reviewer 1 ·

Basic reporting

The paper meets these standards.

Experimental design

The experimental design relates to inclusion and exclusion of studies included in the meta-analysis. It seems that they choose to include non-peer-reviewed papers but did not justify or even comment on this aspect of their decision. Furthermore, there are many studies that were excluded, again without justification.

In addition, the authors sub-divided the studies along category lines but provided no theoretical explanation for these divisions. Given that these sub-divisions were central to the entire paper, this is a serious limitation to the research.

Thus, I do not believe experimental design standards were met.

Validity of the findings

Had the meta-analysis been more complete and the categories been justified, the validity of the findings met standards. This was a strength of the article but ultimately was insufficient due to the points raised above.

Additional comments

The authors conducted a meta-analysis on various studies exploring whether bodily states influence spatial perception. They found high levels of bias and low levels of evidence. They conclude that more reliable measures are needed.

One thing I liked about the paper was the attempt to draw generalizations across a wide range of studies. I also appreciate the amount of work that was put in to gather all the studies and analyze data across them. One major concern was that I was unconvinced by how they sub-divided the studies. The categories were vague and lacked precise definitions, and the authors offered no theoretical motivation for these sub-categories. The sub-categories were also not part of the pre-registration. The categories were numerous, thus leading to small numbers of studies within some of the categories, making it hard to achieve reliable generalizable conclusions.
I was also left wondering why so many studies were excluded from the meta-analysis. For instance, there were many studies on distance-on-hills effect (e.g. Stefanucci et al., 2005) that were not included. The authors included on study on skill (Taylor et al., 2011) but not many others on sports-related skill such as related to softball players (Gray, 2013; Witt & Proffitt, 2005).

My recommendation is to reject the paper. That said, I would encourage the authors to continue with this work but do so in a way that is more theoretically driven (specifically as it relates to subdividing the categories but also with respect to how they define and consider bias) and more inclusive of the full literature.


Major concerns:
• The tactic taken by the authors was to group small sets of studies together based on their classifications of the study parameters and make conclusions about the presence of an effect. A major concern is the vagueness by which action-related concepts were defined.
• For example, how long must a person engage in a strenuous activity for fatigue to be manipulated? How is fatigue defined? Muscle-fatigue? Feelings of fatigue? How does one discriminate between fatigue and warming-up the muscles? Some of the descriptions of fatigue manipulations sound more like what I would do to warm up and prepare for a fatiguing work-out than manipulations of fatigue itself.
• As another example, what should count as “fitness”? How does hip-to-width ratio count? Should self-rated fitness be grouped with other metrics of fitness? Should skills (as in parkour) be counted as fitness? I appreciated the brief discussion of nuance with respect to this topic.
• If the literature is rife with low-powered studies and publication bias (as seems to be the case for much of psychology!), what value is there for a meta-analysis?
• Determining how to sub-divide the categories is a challenging, and theoretically interesting, task. The authors offered no theoretical reason for the divisions they selected, and these divisions were different than the ones they had pre-registered (e.g. virtual vs non-virtual manipulations). I wonder if the researchers can use a more data-driven approach. Perhaps some kind of PCA could be interesting.
• External Load: Wasn’t this already addressed in the Molto paper? What is added here? Were any additional studies added? Were any excluded?
• I am surprised they used studies from non peer-review journal publications (dissertations, conference proceedings). At least, they should indicate which are which when discussing each.
• I found they used “bias” to refer to bias related to the study design (e.g. they refer to sampling bias as the use of convenience sampling) and to bias that could drive systematic results (such as response bias), and this was confusing.
• There were instances for which they argued certain variables should be controlled for (like stress and mood), but they did not provide any evidence that these variables are relevant. It is therefore misleading to tell people to control for them without engaging in more critical thought of the impact of doing such analysis. As a concrete example, some researchers may take the authors’ recommendations as meaning that they should re-analyze their data and include these covariates, but this inflates the chance of a Type I error assuming the researcher is more likely to report only the analyses that resulted in significance (an assumption that has been supported in the literature).

·

Basic reporting

The basic goal of a meta-analysis is to provide an overview of the strength of evidence. A weakness of the method (see Friese & Frankenbach, 2020) is that it tends to amplify effects of publication bias and p-hacking). I think this second point might be profitably acknowledged either in the introduction or the conclusion.

Reference
Friese, M., & Frankenbach, J. (2020). p-Hacking and publication bias interact to distort meta-analytic effect size estimates. Psychological Methods, 25(4), 456.

Experimental design

Overall, I think the approach taken by the authors is within the bounds of currently accepted practice. However, in my view, the design of this meta-analysis appears to be subject to confirmation bias problems. That is, alternative explanations and conflicting findings are not deeply considered. Rather, outcomes are taken at nearly face value. A study that considers the evidence for one theory and ignores evidence for alternative theories is considered a poor design. But with meta-analysis, there is no convenient way to accomplish this. A meta-analysis is really about replicabilty, but does not address the trickier problem of interpretability. I'm not sure this is correctable, but I don't see that it should be held against the authors. In my view the final word of the initial question in the title could equally be "say" instead of "see."

Validity of the findings

In my opinion, the authors seek to be fairly judicious, but the biased focus on whether there is support for embodied perception (rather than how the support for embodied perception compares with the evidence of conflicting theories, such as knowledge effects, or social effects on compliance with experimental demands) limits the value of this paper.

Additional comments

Overall, I believe that the authors have competently applied the method of meta-analysis to the literature on embodied effects on perception. If the primary issue in this literature was replication, this would be a very useful approach. In my view however, the primary problem is interpretation (including the absence of controls). I do not ask that the authors be required to share my views, but I think some comments about the limits of meta-analysis are in order. The authors suggest that future studies must be better about certain forms of bias, but they don't address publication bias at all, which, for some of us appears to be a serious problem with this literature -- a problem which meta-analysis risks amplifying.

Reviewer 3 ·

Basic reporting

Thank you for the opportunity to review this comprehensive and far-reaching meta-analysis, which investigates the influence of bodily state on spatial perception, under the theoretical banner of the Economy of Action Hypothesis.

The authors have undertaken a large amount of work, presenting their findings extremely clearly, in a structured and thoughtful way.

The categories for analysis were well defined and each aspect discussed in depth, with good use of interim summaries to unpack the broader context.

Many steps were taken to ensure the robustness of the review, including pre-registration via the Open Science Framework, well defined search criteria, the inclusion of a Risk of Bias Assessment and a robust method of extracting data, including in the event of extraction from graphical forms.

Comments for consideration.
Introduction.
General: L49-52. A more comprehensive overview of the theoretical background and placement of the Economy of Action Hypothesis is warranted. This is currently brief, with little context given to the multiple interpretations of embodied theories of perception, enactive approaches and the basis of ecological perception. These theories have individual and environment coupling but have different commitments that can be cached out in the introduction to demonstrate EoA differences and the debate that surrounds cognitive penetration of perception. Inference models of perception could be referenced in general, e.g. Helmholtz; Clark, 2013; Friston, 2013).

L57. Economy of Action may not commit to actively discouraging movement but instead outlines a theory by which appropriate action is taken in line with bodily state (engagement or not; explore/exploit).
L64. This is the first mention of embodied perception, and for clarity should be defined in this context, with reference to the theories mentioned above.

General. L72. An acknowledgement of the key debate raised by firestone and Scholl, with regards to post-perceptual influences could be critiqued further.

L81. A clear and well articulated justification for the present work.

Tables and figures and extremely clear and provide excellent level of detail, relevant to the categories investigated. The supplementary data provided is well organised and thorough.

Minor. L252. rewording suggestion to "Blinding was poorly applied"
L302. Remove Hunt et al, from brackets.

Experimental design

The methodology design is thorough and described in its entirety. The authors present a justified extension to a recently publish review under a complementary topic, filling an important gap in both the impact of bodily capacity and importantly the risk of bias.
The calculation for risk of bias its clear and appropriately deployed.

Each category identified is rigorously investigated, and thoughtful discussions of potential confounders are well articulated at each stage.

Minor. Each interim summary could make more use of the risk of bias in describing the impact of the results found.

Validity of the findings

Although a recent review has been published that outlines a similar investigation: "action effects on visual perception." There is appropriate rationale for the further rigorous investigation undertaken here, contributing to the knowledge around the EoA hypothesis.

The statistical approach is robust and justified. Results are pooled appropriately throughout, and considered independently when necessary.

The conclusion drawn is appropriate, with insightful and well-thought through recommendations for implications and future methodologies.

---

## Round 0.2 · accepted · Accept

After reading your revised manuscript, I am satisfied that you have addressed the critiques provided by reviewers in round 1. Below you will see that one of the original reviewers also read the revision and approved of the updates.

This is a controversial area. We are now at a point where folks like you can begin synthesizing the extant literature. Given the scattershot nature of research in embodiment, this is no small feat! I applaud your attempts to find order within this research literature. I hope that your paper will find a large readership.

Reviewer 3 ·

Basic reporting

The authors have addressed the concerns associated with the span of literature incorporated, ensuring that the context is fully articulated. A clear overview of alternate and competing hypotheses are provided, giving an improved balance to the introduction.

Experimental design

The authors have provided further details on inclusion and exclusion criteria, as well as clarifying their sub-grouping process, which deviated from the original proposal.

The addition of risk of bias, confidence and relation to competing theories in the interim summaries is helpful and clear.

Validity of the findings

No comment

Additional comments

I believe that the authors have addressed my concerns, grounding their results more appropriately in the theoretical context. In addition, they have discussed well the importance of accounting for risk of bias in this literature, despite the methodological shortfalls. They have placed these findings well in relation to competing work.